# PIPE: Physics-Informed Position Encoding for Alignment of Satellite Images and Time Series in Typhoon Forecasting

Haobo Li[1]  Eunseo Jung[1]  Zixin Chen[1]  Zhaowei Wang[1]
Yueya Wang[2]  Huamin Qu[1]  Alexis Kai Hon Lau[2]
[1] Department of Computer Science & Engineering
[2] Division of Environment & Sustainability
Hong Kong University of Science and Technology
hliem@connect.ust.hk

## Abstract

Multimodal time series forecasting is foundational in various fields, such as utilizing satellite imagery and numerical data for predicting typhoons in climate science. However, existing multimodal approaches primarily focus on utilizing text data to help time series forecasting, leaving the visual data in existing time series datasets underexplored. Furthermore, it is challenging for models to effectively capture the physical information embedded in visual data, such as satellite imagery's temporal and geospatial context, which extends beyond images themselves. To address this gap, we propose **p**hysics-**i**nformed **p**ositional **e**ncoding (**PIPE**), a lightweight method that embeds physical information into vision language models (VLMs). **PIPE** introduces two key innovations: (1) a physics-informed positional indexing scheme for mapping physics to positional IDs, and (2) a variant-frequency positional encoding mechanism for encoding frequency information of physical variables and sequential order of tokens within the embedding space. By preserving both the physical information and sequential order information, **PIPE** significantly improves multimodal alignment and forecasting accuracy. Through the experiments on the most representative and the largest open-sourced satellite image dataset, **PIPE** achieves state-of-the-art performance in both deep learning forecasting and climate domain methods, demonstrating superiority across benchmarks, including a 12% improvement in typhoon intensity forecasting over prior works.

## 1 Introduction

Time series forecasting plays a crucial role in climate modeling [58]. This task involves modeling temporal dependencies to predict future values of a target variable, a challenge exacerbated by noise, non-stationarity, and the frequent need to integrate heterogeneous auxiliary data. While traditional methods like Autoregressive Integrated Moving Average (ARIMA) rely on statistical priors [12], deep learning architectures (e.g., LSTMs [13], Transformers [47]) have recently dominated the field by learning latent temporal patterns from data. However, these methods still struggle to deliver precise forecasts amid the complexity and scale of real-world data, leaving high-stakes tasks such as typhoon-track prediction continue to have a long way to go.

The rise of large language models (LLMs) as a type of sequence modeling has introduced new opportunities for time series forecasting. Although LLMs were originally built for NLP tasks such as text generation [36] and summarization [7], their core objective naturally aligns with time-series

39th Conference on Neural Information Processing Systems (NeurIPS 2025).

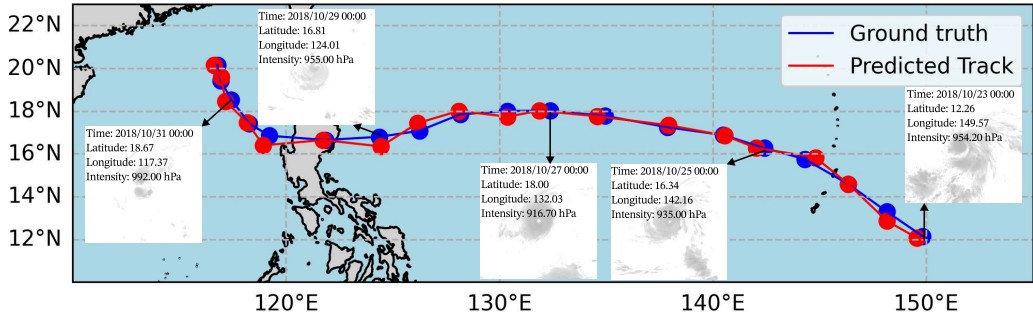

Figure 1: The multimodal time series forecasting task and the forecasting results for Typhoon Yutu by our PIPE-3B. The leading time is 12 hours and the time gap between neighbouring dots is 12 hours. In multimodal time series forecasting, satellite images can improve the forecasting accuracy.

forecasting: predicting the next token in a sentence mirrors forecasting the next value in a sequence, both conditioned on historical context. Consequently, existing work adapts LLMs to forecasting through tokenization techniques or patching technology to splice time-series segments into model context [62]. More recent work broadens the paradigm by injecting auxiliary instructions or descriptions through zero-/few-shot inference [4], in-context learning [32], and text-augmented forecasting [17]. Several studies push the scope further by incorporating explicit temporal cues, for example, TimeLLM [17] and UniTime [29] involve temporal information in prefix-prompts, while AutoTimes embeds timestamps as positional encodings to integrate the temporal information [32].

However, existing methods for multimodal time series forecasting, which integrate visual and numerical data, face numerous limitations. Integrating visual context, such as satellite imagery, into forecasting is indispensable in climate [48] and other domains [15, 52], yet state-of-the-art vision–language models (VLMs) like GPT-4o [36], Gemini [46], and Qwen-VL [2] are tuned primarily for general domain multimodal data. Furthermore, their projection layers, the vision encoder from CLIP [41], cross-attention [27], Q-Former [24], and MLP [28], solely focus on pixel-level semantics and overlook the rich physical metadata (e.g., timestamps and geo-coordinates) embedded in real-world imagery. This omission limits their capacity to improve high-stakes multimodal forecasting tasks. For example, typhoon track prediction with satellite imagery (Figure 1) requires correlating pixel values with the time-specific geophysical attributes (e.g., latitude, longitude) embedded in each pixel. As a result, addressing these overlooked physical dimensions beyond the pixel-level values in multimodal time-series forecasting not only fills a critical gap in existing alignment methods that only focus on the pixel-level values but also introduces a new task for multimodal alignment.

To address these challenges, we propose **p**hysics-**i**nformed **p**ositional **e**ncoding (**PIPE**), a lightweight method to embed latent physical metadata (e.g., timestamps, geospatial coordinates) into positional encodings. Unlike traditional positional encodings, which focus solely on the sequence order of tokens within one input instance [47, 11], **PIPE** encodes shared global physical knowledge (e.g., latitude-longitude relationships consistent across instances) while preserving sequence order information. Specifically, **PIPE** introduces two key innovations: (1) a physics-informed positional indexing scheme that maps physics to positional IDs, and (2) variant-frequency positional encoding that integrates the attributes of physical variables in the input embedding space. They maintain the original token topology while enabling explicit modeling of geographic-temporal dependencies. Experiments on the most representative and the largest open-sourced satellite image dataset for typhoons, Digital Typhoon [22], demonstrate improved cross-modal integration and forecasting accuracy. **PIPE** achieves state-of-the-art performance compared to general AI and domain models across multiple benchmarks on the multimodal time series forecasting task.

Our contributions are threefold:

- We propose the multimodal time-series forecasting scheme to integrate visual information, where time series data is accompanied by corresponding vision data, extending beyond conventional univariate/multivariate time series forecasting.

- We propose the **PIPE**[1], a method to embed physical knowledge into VLMs. Our method contains two key innovations: (1) physics-informed positional indexing and (2) variant-frequency positional encoding.
- Through comprehensive experiments on the most representative task and the largest open-sourced satellite image dataset, we show an obvious gain (12% for intensity forecasting) after appropriately integrating vision and physics. Through the ablation study, we quantify the benefits of (1) integrating visual data for multimodal time series forecasting (8% for intensity forecasting) and (2) integrating physics knowledge (6% for intensity forecasting).

## 2 Related Work

### 2.1 Transformers for Time-series Forecasting

Transformers are widely used for time series forecasting, demonstrating superior performance over traditional statistical models and RNN [42] architectures. Key innovations driving this success include efficient attention mechanisms and architectural adaptations tailored to temporal patterns. Recent works have introduced several enhancements to address computational complexity and domain-specific challenges. Informer [61] tackles the quadratic complexity of standard self-attention through ProbSparse attention combined with distillation operations to prioritize crucial temporal features. Autoformer [57] integrates decomposition from time-series analysis with autocorrelation-inspired attention and outperforms self-attention in both efficiency and accuracy. iTransformer [31] applies the attention and network on the inverted dimensions for time series forecasting. One Fits All [62] fine-tunes on all major types of tasks involving time series. Other variations on transformer include CrossFormer [51], TimeXer [53], TimeMixer [50], etc.

The patching paradigm has inspired multiple variants. PatchTST [34] segments time series into local windows as input tokens while maintaining channel independence for multivariate data. Building on this concept, works such as Pathformer [5] and Sageformer [59] research transformer-based patching technology in terms of multiscale and inter-series dependencies. Notably, works such as One Fits All [62] and Time-LLM [17] demonstrate the transferability of patching strategies by adapting pre-trained large language models to time-series forecasting through input token alignment.

However, these developments underscore the challenge of managing complexity when incorporating additional modules, such as patch-based components. Our method incorporates physical information via Position IDs, avoiding the need for extra models.

### 2.2 Multimodal LLMs for Time Series Forecasting

Recent advances in LLMs have catalyzed efforts to develop multimodal models capable of processing diverse data modalities (e.g., text, images, audio) through unified architectures. This paradigm has inspired time-series forecasting adaptations that integrate textual instructions with temporal data. TimeLLM [17] reprograms the input time series with text timestamps as prefix-prompts to align the two modalities. Unitime [29] utilizes prefix-prompts to encode frequency information of temporal data to augment the model. AutoTimes [32] uses the embedding of textual timestamps as the position encoding to incorporate temporal information. Subsequent works like UrbanGPT [25], TEST [45], ChatTime [49], and GPR4MTS [16] utilize similar methods, aligning text instructions and time series for the augmentation of time series forecasting.

However, for time series forecasting, existing multimodal approaches focus narrowly on aligning textual instructions with numerical time series, neglecting critical vision modalities inherent to many forecasting scenarios, such as typhoon forecasting. Our work researches the utilization of vision data for time series forecasting.

### 2.3 Position Encoding in Transformers

Transformers require explicit position encoding to capture sequential order information, unlike RNNs that inherently model temporal relationships through hidden state propagation. Current position encoding strategies can be categorized into two primary paradigms:

---

[1]https://github.com/hobolee/PIPE

1. Absolute position encodes the absolute position of a unit within a sentence. The original Transformer architecture [47] introduced two variants: 1) Learned positional embeddings during training stages. 2) Fixed sinusoidal functions:

$$PE_{(pos,2i)} = sin(\frac{pos}{10000^{2i/d_{model}}}) \tag{1}$$
$$PE_{(pos,2i+1)} = cos(\frac{pos}{10000^{2i/d_{model}}})$$

where $i$ denotes the dimension, $pos$ is the position, and $d_{model}$ is the dimension of embeddings. This matrix is simply added to the embeddings before they are fed to the Transformer model. Subsequent methods have been proposed to address the challenges of long sequences [20, 30] and improve efficiency [39].

2. Relative position encodes the position of a unit relative to other units. Shaw et al. [43] pioneered this approach by modifying self-attention to compute relative position biases. Transformer-XL [8] introduces recurrence-aware position encoding for long-context modeling. Ke et al. [19] propose untied position embeddings to add relative position embeddings through additive scalar biases. Wu et al. [54] propose to incorporate the real distances between tokens to re-scale the raw self-attention weights. Rotary Position Embedding (RoPE) [44] injects relative positions via rotation matrices.

Though effective for local sequence modeling, these methods focus on intra-instance positional relationships within individual input samples. For time-series forecasting tasks where cross-instance physical dependencies are critical (e.g., all instances share the global knowledge of geographic information), existing approaches fail to capture global temporal-spatial correlations across the entire dataset. Our work addresses this limitation through physics-informed position encoding. By encoding global timestamps with geographic coordinates (latitude/longitude), our method preserves continuous spatiotemporal relationships across independent time-series sequences.

## 3 Method

This section formalizes the multimodal time series forecasting problem and proposes physics-informed positional encoding that integrates physical information into VLMs for multimodal time series forecasting. A schematic overview of the method is provided in Figure 2.

### 3.1 Multimodal Time Series Forecasting Problem Formulation

We address the problem of multimodal time series forecasting, where historical observations comprise both time series data of multiple variables and visual images. Given a sequence of historical time steps:

$$\boldsymbol{x}_{t-H+1:t} = \{\boldsymbol{x}_{t-H+1}, \boldsymbol{x}_{t-H+2}, ..., \boldsymbol{x}_t\} \in \mathbb{R}^{H \times C} \tag{2}$$

where $H$ denotes the historical time steps, $C$ the number of variates, along with a corresponding sequence of $H$ images: $\boldsymbol{i}_{t-H+1:t} \in \mathbb{R}^{3 \times H_{img} \times W_{img}}$ for each time step with $H_{img}, W_{img}$ as the height and width of the image, the objective is to forecast the future $F$ time steps:

$$\boldsymbol{x}_{t+1:t+F} = \{\boldsymbol{x}_{t+1}, \boldsymbol{x}_{t+2}, ..., \boldsymbol{x}_{t+F}\} \in \mathbb{R}^{F \times C} \tag{3}$$

Our task is to propose a VLM model as a cross-modal forecaster $f_{VLM}(\cdot)$ to model cross-modal relationships between the multivariate sequence $\boldsymbol{x}_{t-H+1:t}$ and visual sequence $\boldsymbol{i}_{t-H+1:t}$. Formally, we seek to learn:

$$\hat{\boldsymbol{x}}_{t+1:t+F} = f_{VLM}(\boldsymbol{x}_{t-H+1:t}, \ \boldsymbol{i}_{t-H+1:t}) \tag{4}$$

### 3.2 VLMs for Multimodal Time Series Forecasting

To perform the multimodal time series forecasting, we use the VLM to encode the time series input and the vision input, following the practice of VLM's pipeline. In this paper, we use the open-source Qwen-2.5-vl [2] as it incorporates 3D positional encoding, which enhances its multimodal abilities.

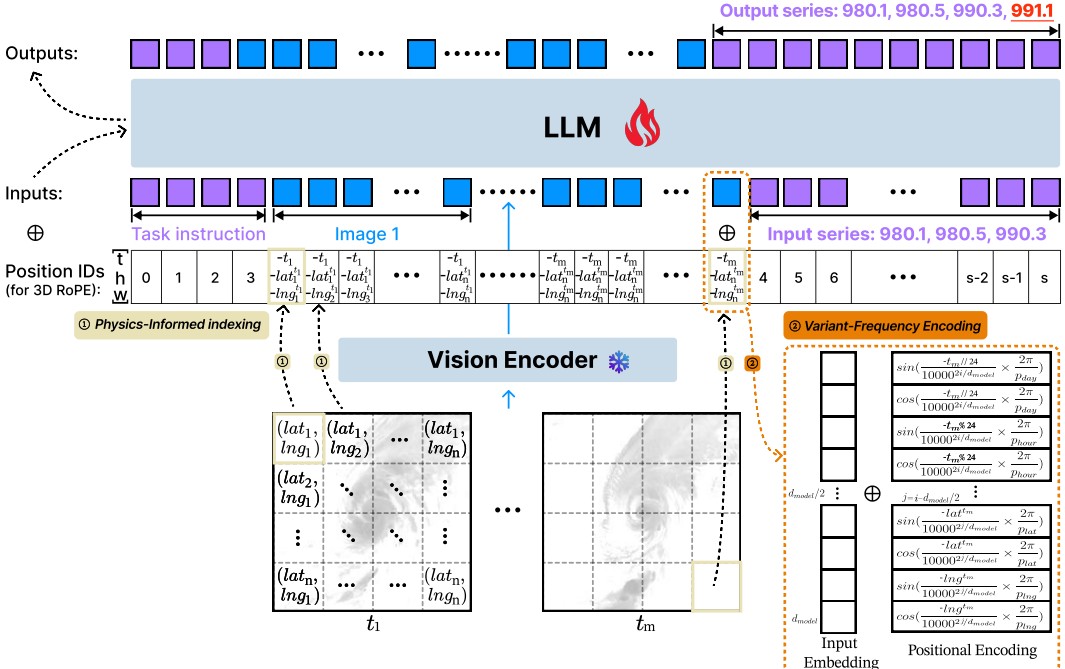

Figure 2: The framework of physics-informed positional encoding. It includes: (1) a physics-informed positional indexing scheme that maps physics to positional IDs, and (2) variant-frequency positional encoding that integrates the attributes of physical variables in the input embedding space.

**Text embedding**  To leverage the capability of the pretrained LLM, we tokenize the time series data, $x_{t-H+1:t}$, into tokens and concatenate them with task-specific instructions (e.g., "Predict next 24 hours of typhoon track"). They are fed into the LLM's transformer layers, as depicted with purple inputs in Figure 2. The LLM is trained during the training stage. The prompt design for multimodal time series forecasting can be found in Appendix A.

**Vision embedding**  Each image $i_t \in \mathbb{R}^{3 \times H_{img} \times W_{img}}$ is split into $N$ non-overlapping patches $\{p_{t,k}\}_{k=1}^{N}$, where $p_{t,k} \in \mathbb{R}^{3 \times 28 \times 28}$. These patches are encoded using the pretrained vision encoder of Qwen-2.5-vl, producing embeddings that are dimensionally consistent with the text tokens, as depicted with blue inputs in Figure 2. The vision encoder is frozen during the training stage.

### 3.3 PIPE

We propose **PIPE** to incorporate physical information into multimodal alignment for multimodal time series forecasting. Our proposed **PIPE** includes two cores: physics-informed positional indexing (Figure 2 ①) and variant-frequency positional encoding (Figure 2 ②). The algorithm can be found in Appendix B.

#### 3.3.1 Physics-Informed Positional Indexing

We propose physics-informed positional indexing to integrate physical information into the model.

**Schemes of indexing position IDs.**  Incorporating physical information into the model using positional IDs provides a direct solution without the need for additional structural complexity. We explore three indexing strategies to facilitate this integration:

(1) Sequential indexing. The most intuitive approach is to follow the standard transformer practice [47] and ViT [11], using the sequence to index the position IDs. In this scheme, the positional IDs are indexed linearly as:

$$position\_ids = [0, 1, 2, \ldots, seq\_len - 1] \tag{5}$$

where $seq\_len$ represents the total length of the input sequence, including both text tokens and vision tokens. This approach effectively encodes 1D sequential order but lacks explicit order information of the image (2D) or video (3D) for multimodal inputs.

(2) 3D indexing. Building on Qwen-2.5-VL [2], this method expands positional indexing to include three independent dimensions: temporal, height, and width, for the alignment of images and videos.

- Text tokens continue to use sequential indexing described in Equation 5, while vision tokens are indexed based on their temporal and spatial attributes.

- Temporal positions of vision tokens are calculated as:

$$t = tokens\_per\_second \times temporal\_patch\_size/fps \qquad (6)$$

  where $tokens\_per\_second$ dictates how many time steps are conceptually packed into a one-second interval of the video, $temporal\_patch\_size$ is the number of frames, and $fps$ is the video's frame rate.

- For spatial dimensions of vision tokens, the height and width positional IDs correspond to a patch grid ranging from $(0, 0)$ to $(N_{row} - 1, N_{col} - 1)$, where $N_{row}$ and $N_{col}$ are the numbers of image patching in height and width, respectively. Although this 3D indexing scheme aligns temporal and spatial order within vision tokens, it only captures the intra-relationship of positions with the input instance. It does not explicitly encode extra-physical properties such as time, latitude, and longitude, which are global knowledge among all instances in the dataset.

(3) Physics-Informed positional indexing (Figure 2). To address the limitations of 3D indexing, we propose a novel physics-informed positional indexing scheme that explicitly integrates global knowledge of physical attributes into positional IDs.

- Text embeddings continue to use the sequence indexing scheme described in Equation 5.

- Temporal positional IDs of vision tokens are computed based on the hourly progression of a given year. Specifically, the temporal position ID is calculated by:

$$t = t_{day} \times 24 + t_{hour} \qquad (7)$$

  where $t_{day}$ is the day of the year (ranging from 0 to 365) and $t_{hour}$ represents the hour of the day (ranging from 0 to 23). This indexing introduces meaningful temporal patterns aligned with real-world time progression.

- The height and width positional IDs of vision tokens are determined using the latitude and longitude of the image patch centers.

To prevent the performance decreases caused by the conflicts between the physical information of vision tokens and the order information of text tokens (refer to the ablation experiment section 4.6), we map the range of vision positional IDs ($t$: $0 - 8784$ (8784 hours in a year), $lat$: $0 - 180$, $lng$: $0 - 360$) to negative values. This avoids overlap with the text positional ID range, ensuring smooth multimodal integration. Moreover, temporal, latitudinal, and longitudinal dimensions are inherently independent, eliminating concerns about overlap.

After incorporating the cross-instance physical information among all input samples using physics-informed positional indexing, we apply RoPE [44] on position IDs to encode intra-instance positional relationships within individual input samples.

### 3.3.2 Variant-Frequency Positional Encoding

We also merge the information of the physical variables into input embeddings. To differentiate between physical variables, we modify the standard sinusoidal positional encoding (Equation 1) by introducing a variant-frequency sinusoidal function.

**Variant-frequency sinusoidal function** This function modifies the sine and cosine components and the target dimension based on the temporal, latitude, and longitude frequencies. Figure 2 illustrates the setting, Equation 9 in the Appendix gives the complete definition, and Figure 4 visualizes

the function. For conciseness, the function for image tokens can be formulated as:

$$PE_{(pos,2i)} = sin(\frac{pos}{10000^{2i/d_{model}}} \times \frac{2\pi}{p})$$

$$PE_{(pos,2i+1)} = cos(\frac{pos}{10000^{2i/d_{model}}} \times \frac{2\pi}{p})$$

(8)

where $pos$ can be $t_{day}$, $t_{hour}$, $lat$, and $lng$ depending on the dimensions. $t_{day}$ is the day of the year (ranging from 0 to 365) and $t_{hour}$ represents the hour of the day (ranging from 0 to 23). $lat$ is the latitude of the image token, and $lng$ is the longitude of the image token. $p$ represents the wavelength specific to physics. For temporal data, $p_{day} = 366$ and $p_{hour} = 24$ and for spatial dimensions, $p_{latitude} = 180$ and $p_{longitude} = 360$. After the modification, the wavelengths form a geometric progression from $p$ to $p \cdot 10000/2$ for vision data.

Text tokens preserve the standard sinusoidal encoding to maintain compatibility with pretrained LLM structures. These variant-frequency position encodings are added to the input embeddings at the bottom of the decoder stacks after they are divided by $d_{model}$. They map different physical variables to distinct frequency domains before incorporating them into the input embedding space.

# 4 Experiments

This section presents a systematic evaluation of the proposed method for the most representative multimodal time series forecasting task, typhoon forecasting. We first describe the datasets, baseline methods, and evaluation metrics, followed by the experiments and ablation studies.

## 4.1 Dataset

For multimodal time series forecasting, we utilize the open-source Digital Typhoon dataset [22], the longest hourly satellite imagery collection dedicated to typhoon analysis spanning 40+ years (1978–2023) with a 5 $km$ spatial resolution. The spatial coverage of the dataset is the Western North Pacific basin. The dataset includes 1,116 typhoon sequences and 192,956 images (resolution of 512×512 and resized to 224x224). The size of the dataset is different from the size in the original paper since the dataset is being regularly updated.

Typhoon track annotations, including intensity, latitude, and longitude, are sourced from the Best Track dataset [23]. It is the best estimate, a globally recognized benchmark derived from retrospective post-event analysis. This metadata ensures reliable spatiotemporal grounding, as it synthesizes all available observational data to reconstruct each typhoon's lifecycle with high precision. In our experiments, we will forecast three variables: intensity, latitude, and longitude. The dataset is split using a ratio of 0.7:0.15:0.15 based on the typhoon sequences as the original dataset.

## 4.2 Baselines

**Domain models** For typhoon forecasting, we compare our method against the state-of-the-art domain-specific NWP-based model: forecasting system of the European Centre for Medium-Range Weather Forecasts (ECMWF) [10] and two environment-domain large models, Pangu [3] and Gen-Cast [40], which serve as domain-specific benchmarks. Additionally, we include comparisons with the domain practice method, Typhoon Intensity Forecasting based on the SHIPS method (TIFS) [35]. We report only the available performance from their paper and do not retrain the models, as we cannot reproduce these domain models.

**AI models** We train the state-of-the-art AI models with our dataset, including Transformer-based models (PatchTST [34], iTransformer [31], Crossformer [51], TimeXer [53]) and linear-based models (TiDE [9]), LLM-based model (One Fits ALL [62], AutoTimes [32]), and other models (Times-Net [55], TimeMixer [50]). Due to their model design, they do not incorporate visual data. For the visual data integration, we include benchmark results reported in the original dataset publication (only the leading time of 12h is available) [22], closed-source Gemini-2.5-flash [6] (zero-shot), and train the original Qwen-2.5-VL [2].

Table 1: Multimodal time series forecasting results (leading time is 6h).

| | Models | Intensity (hPa) | | Latitude (°) | | Longitude (°) | | Distance (km) |
|---|---|---|---|---|---|---|---|---|
| | | MAE | RMSE | MAE | RMSE | MAE | RMSE | MAE |
| domain | ECMWF-HRES [10] | | | | | | | 27.181 |
| | PanGu [3] | | \ | | \ | | \ | 32.892 |
| | GenCast [40] | | | | | | | 20.331 |
| | TIFS [35] | \ | 7.292 | | | | | \ |
| w/o vision | PatchTST [34] | 1.806 | 2.867 | 0.199 | 0.266 | 0.322 | 0.404 | 44.537 |
| | iTransformer [31] | 1.848 | 2.979 | 0.164 | 0.231 | 0.203 | 0.281 | 31.248 |
| | Crossformer [51] | 2.389 | 3.599 | 0.310 | 0.418 | 0.520 | 0.684 | 71.216 |
| | TimeXer [53]) | 3.037 | 4.523 | 0.306 | 0.411 | 0.411 | 0.538 | 59.720 |
| | TiDE [9] | 1.724 | **2.819** | 0.161 | 0.224 | 0.237 | 0.312 | 34.068 |
| | One Fits All [62]) | 1.849 | 2.976 | 0.170 | 0.239 | 0.211 | 0.290 | 32.450 |
| | AutoTimes [32] | 1.991 | 3.088 | 0.190 | 0.265 | 0.279 | 0.364 | 40.036 |
| | TimesNet [55] | 2.401 | 3.711 | 0.465 | 0.630 | 0.855 | 1.124 | 113.718 |
| | TimeMixer [50] | 1.913 | 2.973 | 0.177 | 0.237 | 0.238 | 0.313 | 35.374 |
| vision | Gemini-2.5-flash [6] | 1.924 | 3.654 | 0.174 | 0.254 | 0.237 | 0.798 | 36.006 |
| | Qwen-2.5-VL-3B [2] | 1.617 | 3.231 | 0.087 | 0.162 | 0.103 | 0.187 | 17.129 |
| | **PIPE-3B** | **1.515** | 2.981 | **0.084** | **0.159** | **0.095** | **0.178** | **16.275** |

**Implementation Details**   Both our method and baselines use the same temporal settings with the same length of input and output sequences (12h). For One Fits All and AutoTimes, we use their official implementations. Other models without vision, their implementations are through the publicly available Time-Series-Library [56]. For the Qwen-2.5-VL model and **PIPE**, we use LLama-Factory [60] for their implementation. More implementation specifics, including hyperparameters and training protocols, are detailed in Appendix C.

## 4.3   Evaluation Metrics

In NLP tasks, metrics such as ROUGE [26] and BLEU [37] are commonly employed as the metrics. In our cases, we focus on the numerical output. Specifically, for forecasting intensity, latitude, and longitude, we use Root Mean Square Error (RMSE) and Mean Absolute Error (MAE) as primary metrics. When the model accurately predicts these numerical values based on satellite images, we consider it to have effectively aligned the satellite imagery with the time series data. Additionally, we use geographiclib [18] to calculate the position error of typhoon tracks based on the latitude and longitude, following the domain practice.

## 4.4   Main Results

Short-term forecasting plays a crucial role in enabling timely decision-making. Therefore, the forecasting performance of multimodal time series models is presented for lead times of 6 hours (Table 1) and 12 hours (Table 5). The best results are emphasized in bold, while the second-best results are marked with underline.

Overall, our method achieves state-of-the-art performance across the majority of evaluation metrics, demonstrating the efficacy of the proposed **PIPE** in integrating physical information during multimodal alignment. For the 6-hour lead time (Table 1), our model outperforms baselines in most metrics. For example, it shows 12% improvement of MAE for typhoon intensity forecasting when compared to the best w/o vision models TiDE. The sole exception is the RMSE for intensity forecasting, where TiDE and PatchTST exhibit marginally superior performance. These results show the effectiveness of our approach. A critical observation is the consistent superiority of models incorporating vision data over unimodal alternatives. This finding emphasizes the importance of leveraging multimodal inputs to enhance forecasting accuracy in complex spatiotemporal tasks.

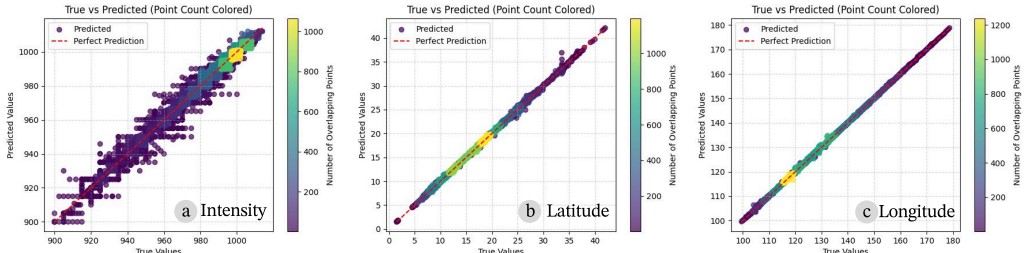

Figure 3: The visualization for the regression results between predicted values and true values (leading time is 6 hours). (a) Plots for intensity. (b) Plots for latitude. (c) Plots for longitude.

## 4.5 Regression Analysis

The regression results of all test typhoon sequences (Figure 3) demonstrate that our method achieves accurate typhoon predictions. Notably, the model exhibits superior performance in location forecasting compared to intensity forecasting, which may be attributed to the richer spatial information provided by satellite imagery for tracking movement. Additionally, the model shows better predictive performance when typhoon intensity is weaker (i.e., higher central pressure, around $1000\ hpa$).

## 4.6 Ablation Study

The results of the ablation study are presented in Table 2 and Table 6, with the best performance highlighted in **bold**. We systematically evaluate three critical components: vision inclusion, physics-informed position indexing, and variant-frequency sinusoidal function.

**The gain of aligning vision**    The inclusion of satellite vision data yields significant improvements in forecasting accuracy. Specifically, the MAE for intensity forecasting improves by up to 8% when the leading time is set to 6 hours. This demonstrates that cross-modal learning effectively leverages spatial patterns in satellite imagery to complement time series data.

**Comparison of schemes of indexing position IDs**    Our physics-informed indexing scheme addresses the critical challenge of preserving physical knowledge while avoiding token order conflicts. To assess its effectiveness, we compare different schemes for indexing position IDs. Specifically, we evaluate the performance by (a) removing the 3D indexing scheme (replacing it with sequential indexing), and (b) removing physics-informed indexing while retaining the 3D indexing scheme. The results show that while sequential indexing and 3D indexing perform similarly, both exhibit a noticeable performance degradation (6% for MAE of intensity forecasting) compared to the physics-informed indexing scheme. Avoiding the overlap between the physical information of vision tokens and the order information of text tokens is critical. There is a dramatic performance decrease when they share overlapping ranges (e.g., longitude: $0-360$ and text tokens: $0-seq_{len}$ ($seq_{len}$ is the number of text tokens)). By mapping the position IDs of vision tokens to negative values, we preserve the physical information and resolve such conflicts, leading to improved performance.

**The gain of integrating physical variables' frequency**    The incorporation of frequency characteristics of physical variables improves physical variable modeling. We show the importance by removing the entire sinusoidal function and only removing the variant-frequency sinusoidal function. The results reveal that our designed sinusoidal function plays a crucial role in aligning the model with the frequency information of physical variables. Its inclusion enhances the model's ability to leverage these variables effectively, leading to improved performance.

**The necessity of combining physics-informed indexing and variant frequency positional embeddings**    In terms of track error, PIPE achieves an error of 16.275, compared to 16.860 when using only physics-informed indexing, 17.177 when using only variant frequency positional embeddings, and 17.129 for the baseline. These results clearly demonstrate that the combination of both components delivers the most significant performance improvement.

Table 2: The results of the ablation study (leading time is 6h).

| Models | Intensity (hPa) | | Latitude (°) | | Longitude (°) | | Distance (km) |
| | MAE | RMSE | MAE | RMSE | MAE | RMSE | MAE |
|---|---|---|---|---|---|---|---|
| w/o vision | 1.646 | 3.220 | 0.088 | 0.160 | 0.102 | 0.193 | 17.235 |
| w/o 3D indexing (using sequence) | 1.628 | 3.749 | 0.087 | 0.163 | 0.102 | 0.185 | 17.084 |
| w/o physics-informed indexing (using 3D) | 1.617 | 3.231 | 0.087 | 0.162 | 0.103 | 0.187 | 17.129 |
| w/o negative indexing | 1.961 | 3.926 | 0.206 | 0.360 | 0.388 | 0.674 | 53.548 |
| w/o entire sinusoidal function | 1.545 | 3.053 | 0.085 | **0.157** | 0.097 | 0.180 | 16.554 |
| w/o variant-frequency sinusoidal function | 1.639 | 3.178 | 0.086 | 0.161 | 0.101 | 0.183 | 16.860 |
| w/o entire physics-informed indexing | 1.581 | 2.994 | 0.087 | 0.161 | 0.102 | 0.186 | 17.117 |
| **PIPE-3B** | **1.515** | **2.981** | **0.084** | 0.159 | **0.095** | **0.178** | **16.275** |

Every component contributes to the multimodal time series forecasting, with vision alignment providing complementary visual patterns, the physics-informed indexing scheme ensuring physical knowledge integration, and the variant-frequency sinusoidal function incorporating physical variables' frequency information.

## 4.7   Generalizability Experiment

To demonstrate the generalizability of our method, we add an additional experiment in the Australian region (AU) [21]. The settings remain the same as the main experiment. The results are shown in Table 3. The results demonstrate that our method can be applied to various regions for typhoon forecasting. Even the model trained on the West Pacific region can perform well in the new dataset (zero-shot). For instance, compared to Qwen-2.5-VL-3B (after tuning), PIPE-3B (Zero-shot) achieved a lower intensity RMSE (2.773 vs. 2.806) and a smaller distance error (19.090 vs. 19.718). Additionally, PIPE-3B (after tuning) outperformed both models, achieving the best performance.

Table 3: The results of the AU Typhoon Experiment (leading time is 6h).

| Models | Intensity (hPa) | | Latitude (°) | | Longitude (°) | | Distance (km) |
| | MAE | RMSE | MAE | RMSE | MAE | RMSE | MAE |
|---|---|---|---|---|---|---|---|
| Qwen-2.5-VL-3B (after tuning) | 1.399 | 2.806 | 0.098 | 0.201 | 0.121 | 0.211 | 19.718 |
| PIPE-3B (Zero-shot) | 1.361 | 2.773 | 0.094 | 0.171 | 0.118 | 0.206 | 19.090 |
| PIPE-3B (after tuning) | **1.352** | **2.558** | **0.093** | **0.163** | **0.116** | **0.202** | **18.835** |

# 5   Limitation

While the integration of satellite imagery improves forecasting accuracy, it increases the computational complexity needed to process high-resolution images. To address these limitations, future work will focus on improving the efficiency of integrating vision data into forecasting models to enable longer input sequences and extended forecasting horizons for VLMs. Furthermore, we will explore the incorporation of physical laws or constraints. Beyond embedding physical information, integrating domain-specific physical principles or environmental constraints could improve the model's interpretability and robustness.

# 6   Conclusion

This paper proposes a multimodal time series forecasting task and addresses the challenge brought by integrating satellite imagery. Existing approaches only focus on pixel-level features, overlooking the rich temporal and geophysical context embedded within vision data. We propose **p**hysics-**i**nformed **p**osition **e**ncoding (**PIPE**). Experimental results demonstrate that **PIPE** achieves state-of-the-art performance across multiple benchmarks. Ablation studies further validate the distinct contributions of each component. Future work will explore the integration of additional physical domain knowledge, such as physical laws and constraints, to enhance real-world applicability.

## Acknowledgements

This project is partially supported by RGC TRS grant T22-607/24N and a grant from the Research Grants Council of the Hong Kong Special Administrative Region, China (Project No. RMGS20RG01).

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

## A  Prompts Design

We show our prompt design for the multimodal time series forecasting, taking one instance of Typhoon Yutu as an example.

---

**System Prompt & Task Instruction**

You are a typhoon forecasting expert. Below are the past 12 hours of typhoon data and the corresponding satellite images. Your task is to forecast the hourly data of the typhoon for the next 12 hours, providing the forecast latitude, longitude, pressure in the same format as the past data format.

---

**Past Data**

The corresponding satellite images are: <image> <image> <image> <image> <image> <image> <image> <image> <image> <image> <image> <image>. The historical hourly data from 2018-10-23 01:00:00 to 2018-10-23 12:00:00 is {latitude: [11.65, 11.7, 11.75, 11.8, 11.85, 11.9, 11.95, 11.99, 12.04, 12.09, 12.14, 12.2], longitude: [151.61, 151.41, 151.2, 150.99, 150.79, 150.6, 150.42, 150.26, 150.11, 149.97, 149.83, 149.7], pressure: [974.2, 973.3, 972.5, 971.7, 970.8, 970.0, 967.5, 965.0, 962.5, 960.0, 957.5, 955.0]}.

---

**Label Data**

The forecast hourly data is: {latitude: [12.26, 12.34, 12.42, 12.5, 12.6, 12.7, 12.81, 12.93, 13.05, 13.17, 13.29, 13.4], longitude: [149.57, 149.44, 149.31, 149.18, 149.04, 148.9, 148.76, 148.61, 148.46, 148.31, 148.15, 148.0], pressure: [954.2, 953.3, 952.5, 951.7, 950.8, 950.0, 945.8, 941.7, 937.5, 933.3, 929.2, 925.0].}

---

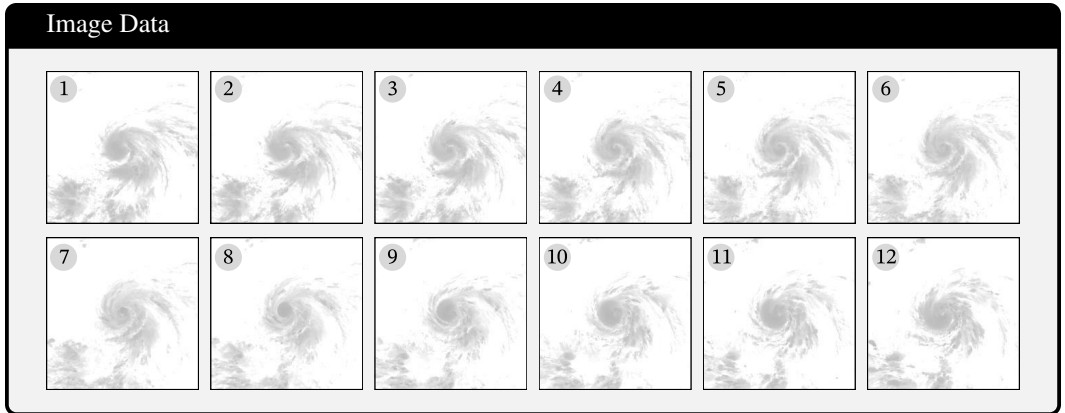

## B Method

### B.1 Algorithms

In this section, we present the algorithms for PIPE (algorithm 1), which integrates physical information into VLMs. To begin, we extract the required physical information (algorithm 2) from the time series data corresponding to each satellite image. This information includes the timestamp and geocoordinates for the typhoon's eye. Additionally, since satellite images are divided into patches, we calculate the geocoordinates for the center of each patch. Next, we incorporate this physical information into the positional encoding of VLMs through physics-informed positional indexing (algorithm 3). Beyond indexing, we adapt the sinusoidal function by introducing variant-frequency sinusoidal encoding (algorithm 4), which embeds the frequency attributes of the variables into the positional embedding. This enhanced positional embedding is then added to the input embedding of the corresponding image tokens. Finally, these integrations are utilized to predict the next token, enabling the model to leverage both spatial and physical context effectively.

---

**Algorithm 1:** PIPE

---

**Require:** Time series input $x$, corresponding image $i$
**Ensure:** Next token prediction $T_{next}$
1: $T_{text}, T_{image}$ = tokenizer($x$), vision_encoder($i$)  ▷ Tokenization
2: $t, lat, lng \leftarrow$ get_physic($x, T_{image}$)  ▷ Extract physical info (algorithm 2): $t, lat, lng \in \mathbb{R}^{1 \times len(T_{image})}$
3: $ids \leftarrow$ position_indexing($T_{text}, T_{image}, t, lat, lng$)  ▷ Compute physics-informed indices (algorithm 3): $ids \in \mathbb{R}^{3 \times len(T_{text}+T_{image})}$
4: $PE \leftarrow$ vf_fun($pos, i$)  ▷ Generate variant-frequency position embedding (algorithm 4): $PE \in \mathbb{R}^{len(T_{text}+T_{image}) \times d_{model}}$
5: $IE \leftarrow [T_{text}, T_{image}] \oplus PE/d_{model}$  ▷ Update input embeddings
6: $T_{next} \leftarrow f_{VLM}(IE, ids)$  ▷ Predict next token

---

**Algorithm 2:** Extract Physical Information for Image Tokens

---

**Require:** Input $x, T_{image}$
**Ensure:** Time ($t_{day}, t_{hour}$) and location ($lat, lng$) for each image tokens
1: $t \leftarrow x$  ▷ Extract temporal information from time series input (Equation 7)
2: $t_{day}, t_{hour} \leftarrow t//24, t\%24$
3: $lat_{image}, lng_{image} \leftarrow x$  ▷ Extract spatial information for the entire image.
4: $lat, lng \leftarrow$ get_center($T_{image}, lat_{image}, lng_{image}$)  ▷ Compute center coordinates for patches

---

**Algorithm 3:** Physics-Informed Positional Indexing

**Require:** $T_{text}, T_{image}, t, lat, lng$
**Ensure:** Physics-Informed $ids$
1: $ids_{text} \leftarrow$ sequential_indexing      ▷ Assign sequential indices to text tokens (Equation 5)
2: $ids_{image} \leftarrow$ physics-informed indexing      ▷ Assign $[t, lat, lng]$ to image tokens (Figure 2)
3: $ids \leftarrow [ids_{text}, ids_{image}]$      ▷ $ids \in \mathbb{R}^{3 \times len(T_{text} + T_{image})}$

---

**Algorithm 4:** Variant-Frequency Sinusoidal Encoding

**Require:** Position $pos$
**Ensure:** Variant-frequency position embedding $PE$
1: $PE_{text} \leftarrow$ standard sinusoidal function (Equation 1)
2: $PE_{image} \leftarrow$ variant-frequency sinusoidal function (Equation 9)
3: $PE \leftarrow [PE_{text}, PE_{image}]$

---

## B.2 Variant-frequency sinusoidal function

This section presents the complete formal definition of the variant-frequency sinusoidal function. The model dimensions are partitioned into two distinct components: temporal dimensions (first half) and spatial dimensions (latter half). Regarding the temporal dimensions, they combine the encoding of $t_{day}$ and $t_{hour}$. Similarly, for the spatial dimensions, they combine the latitude embeddings for $lat$ and the longitude embeddings for $lng$. This dimensional combination enables simultaneous representation of both temporal and spatial characteristics within the unified model framework. We also visualize the function (Figure 4) taking the $d_{model} = 128$ as an example.

$$PE_{(pos,4i)} = sin(\frac{t_{day}}{10000^{4i/d_{model}}} \times \frac{2\pi}{p_{day}}) \; if \; 4i \leq \frac{d_{model}}{2} \tag{9}$$

$$PE_{(pos,4i+1)} = cos(\frac{t_{day}}{10000^{4i/d_{model}}} \times \frac{2\pi}{p_{day}}) \; if \; 4i+1 \leq \frac{d_{model}}{2}$$

$$PE_{(pos,4i+2)} = sin(\frac{t_{hour}}{10000^{4i/d_{model}}} \times \frac{2\pi}{p_{hour}}) \; if \; 4i+2 \leq \frac{d_{model}}{2}$$

$$PE_{(pos,4i+3)} = cos(\frac{t_{hour}}{10000^{4i/d_{model}}} \times \frac{2\pi}{p_{hour}}) \; if \; 4i+3 \leq \frac{d_{model}}{2}$$

$$PE_{(pos,4i)} = sin(\frac{lat}{10000^{4i/d_{model}-1/2}} \times \frac{2\pi}{p_{lat}}) \; if \; \frac{d_{model}}{2} < 4i \leq d_{model}$$

$$PE_{(pos,4i+1)} = cos(\frac{lat}{10000^{4i/d_{model}-1/2}} \times \frac{2\pi}{p_{lat}}) \; if \; \frac{d_{model}}{2} < 4i+1 \leq d_{model}$$

$$PE_{(pos,4i+2)} = sin(\frac{lng}{10000^{4i/d_{model}-1/2}} \times \frac{2\pi}{p_{lng}}) \; if \; \frac{d_{model}}{2} < 4i+2 \leq d_{model}$$

$$PE_{(pos,4i+3)} = cos(\frac{lng}{10000^{4i/d_{model}-1/2}} \times \frac{2\pi}{p_{hour}}) \; if \; \frac{d_{model}}{2} < 4i+3 \leq d_{model}$$

where for temporal dimensions $p_{day} = 366$ and $p_{hour} = 24$, while for spatial dimensions, $p_{latitude} = 180$ and $p_{longitude} = 360$. $t_{day}$ is the day of the year (ranging from 0 to 365) and $t_{hour}$ represents the hour of the day (ranging from 0 to 23). $lat$ is the latitude of the image token, and $lng$ is the longitude of the image token.

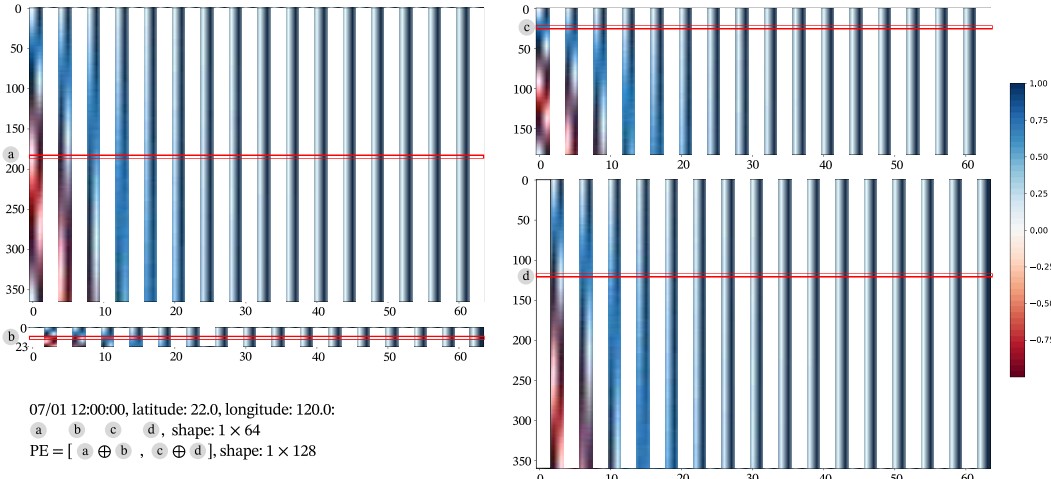

Figure 4: The 64-dimensional positional encoding for the physical variables. Each row represents the embedding vector. The final position encoding will be 128-dimensional by combining them.

## C   Implementation Details

**VLMs training**   We leverage LLama-Factory [60] for training VLMs, utilizing PyTorch [38] on NVIDIA H800 GPUs. The optimization process employs the AdamW optimizer [33] with an initial learning rate of $10^{-5}$ (using the cosine scheduler), a batch size of 1, and CrossEntropy loss over 1 training epoch. We provide the code for the reproduction.

**Non-vision models**   One Fits All [62] and AutoTimes [32] are implemented using their official repositories, adapted to accommodate our typhoon sequence dataset via modifications to the data loader. Configuration follows original specifications: model dimensions of 768 (One Fits All) and 512 (AutoTimes), with a batch size of 128, learning rate of $10^{-4}$, and 10 training epochs. For other AI models, we use the publicly available platform Time-Series-Library [56] to implement them. The parameters for model dimensions and number of heads are based on their implementation (512) with a batch size of 128 and a learning rate of $10^{-4}$, training epochs of 30, and patience of 10.

For domain models, the results are provided by the original paper.

**Training Cost**   We use 4×NVIDIA H800 GPUs to train the models for one epoch. The training time varies significantly across model sizes:

- PIPE-3B: 2.1 hours

- PIPE-7B: 3.7 hours

- PIPE-32B (LoRA [14] rank as 8): 0.7 hours

The PIPE-32B variant achieves substantial time efficiency through LoRA, which reduces trainable parameters while maintaining competitive performance (as shown in Tables 9 and 10). This demonstrates an effective balance between model capacity and computational overhead. For baseline AI models (including LLM-based variants like AutoTimes (OPT model)), training completes in 1 hour with a single NVIDIA RTX 4090 GPU.

## D   Dataset

We provide a comprehensive summary of the Digital Typhoon dataset [22].

Table 4: The detailed information of the Digital Typhoon dataset.

| | Digital Typhoon dataset |
|---|---|
| Temporal coverage | 1978-2023 (present) |
| Temporal resolution | one hour |
| Target satellites | Himawari |
| Spatial coverage | Western North Pacific basin |
| Spatial resolution | 5km |
| Image coverage | 512×512 pixels (1250km from the center) |
| Spectral coverage | infrared (others on the Website) |
| Map projection | Azimuthal equal-area projection |
| Calibration | Recalibration |
| Data format | HDF5 |
| Best track | Japan Meteorological Agency |
| Dataset browsing | Digital Typhoon website |

Table 5: Multimodal time series forecasting results (leading time is 12h).

| | Models | Intensity (hPa) MAE | Intensity (hPa) RMSE | Latitude (°) MAE | Latitude (°) RMSE | Longitude (°) MAE | Longitude (°) RMSE | Distance (km) MAE |
|---|---|---|---|---|---|---|---|---|
| domain | ECMWF-HRES [10] | | | | | | | 44.972 |
| | PanGu [3] | | \ | | \ | | \ | 44.630 |
| | GenCast [40] | | | | | | | **37.930** |
| | TIFS [35] | \ | 9.061 | | | | | \ |
| w/o vision | PatchTST [34] | 3.917 | **5.989** | 0.465 | 0.615 | 0.751 | 0.931 | 103.818 |
| | iTransformer [31] | 4.004 | 6.157 | 0.412 | 0.558 | 0.565 | 0.736 | 83.174 |
| | Crossformer [51] | 4.257 | 6.303 | 0.546 | 0.726 | 0.844 | 1.109 | 118.748 |
| | TimeXer [53]) | 5.380 | 7.911 | 0.563 | 0.755 | 0.713 | 0.962 | 108.665 |
| | TiDE [9] | 3.926 | 6.080 | 0.416 | 0.561 | 0.677 | 0.850 | 93.570 |
| | One Fits All [62]) | 4.039 | 6.212 | 0.420 | 0.568 | 0.586 | 0.759 | 85.555 |
| | AutoTimes [32] | 4.086 | 6.220 | 0.448 | 0.600 | 0.692 | 0.872 | 97.244 |
| | TimesNet [55] | 4.798 | 7.220 | 0.892 | 1.133 | 1.796 | 2.147 | 230.376 |
| | TimeMixer [50] | 4.227 | 6.290 | 0.400 | 0.533 | 0.524 | 0.685 | 78.569 |
| vision | Original paper [22] | \ | 12.100 | \ | | \ | | \ |
| | Qwen-2.5-VL-3B [2] | 3.963 | 6.599 | 0.371 | 0.535 | 0.435 | 0.610 | 69.959 |
| | **PIPE-3B** | **3.855** | 6.333 | **0.359** | **0.526** | **0.411** | **0.587** | 67.114 |

# E  Supplementary Results

## E.1  Forecasting Results of More Leading Times

We present additional forecasting analyses in this section. First, we list the 12-hour lead-time forecasting performance of all baseline models (Table 5). Our model demonstrates state-of-the-art results across the majority of the evaluation metrics. Second, we list the result of the ablation study when the leading time is 12 hours. The consistency between results at different leading times confirms the robustness of our architectural design, demonstrating that all modules contribute meaningfully to forecasting accuracy. Finally, we visualize the MAE for the forecasting of pressure, latitude, longitude, and distance across lead times ranging from 1 to 12 hours (Figure 5). The results confirm that our model consistently achieves the lowest MAE values at all forecast leading times. This systematic advantage over baseline models highlights the effectiveness of our model in maintaining forecasting precision as the leading time increases.

Table 6: The results of the ablation study (leading time is 12h).

| Models | Intensity (hPa) | | Latitude (°) | | Longitude (°) | | Distance (km) |
|---|---|---|---|---|---|---|---|
| | MAE | RMSE | MAE | RMSE | MAE | RMSE | MAE |
| w/o vision | 4.120 | 6.820 | 0.372 | 0.532 | 0.436 | 0.616 | 70.138 |
| w/o 3D indexing (using sequence) | 3.936 | 6.809 | 0.366 | 0.535 | 0.434 | 0.611 | 69.382 |
| w/o physics-informed indexing (using 3D) | 3.963 | 6.599 | 0.371 | 0.535 | 0.435 | 0.610 | 69.959 |
| w/o negative indexing | 4.282 | 7.017 | 0.550 | 0.806 | 0.869 | 1.306 | 124.329 |
| w/o entire sinusoidal function | **3.827** | 6.387 | 0.364 | 0.527 | 0.416 | 0.590 | 67.904 |
| w/o variant-frequency sinusoidal function | 4.071 | 6.689 | 0.370 | 0.537 | 0.429 | 0.604 | 69.389 |
| **PIPE-3B** | 3.855 | **6.333** | **0.359** | **0.526** | **0.411** | **0.587** | **67.114** |

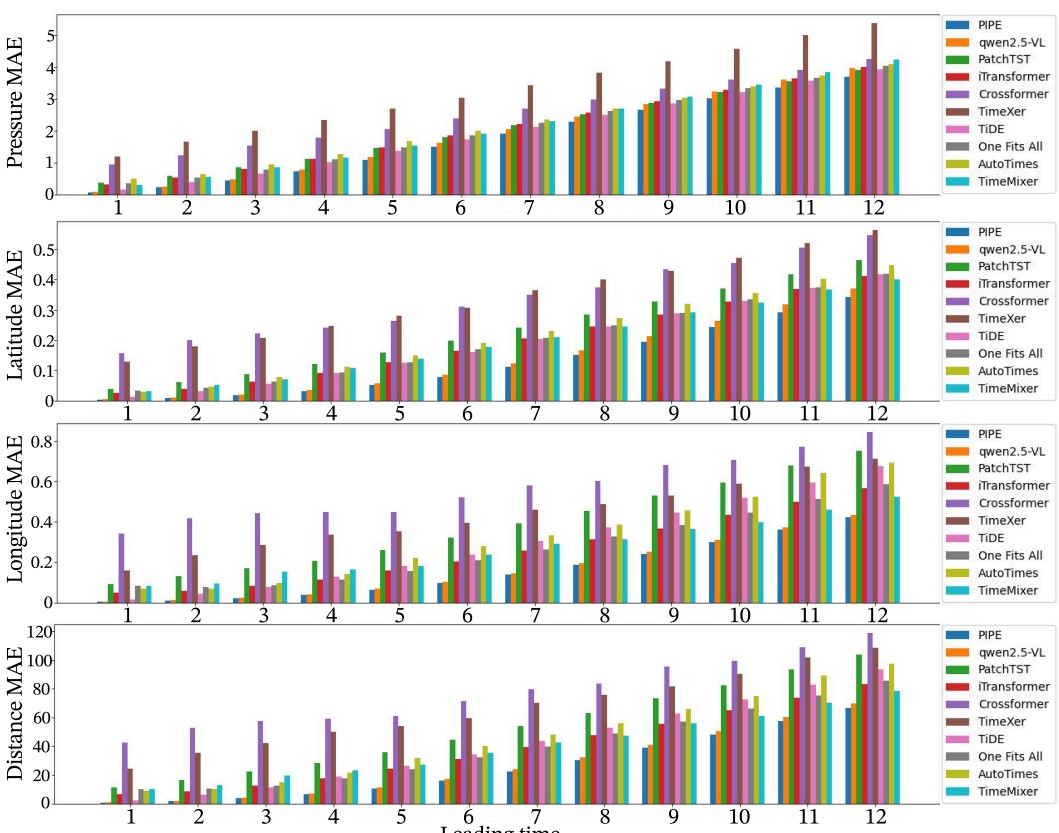

Figure 5: The performance across leading times ranging from 1 to 12 hours.

## E.2 Experiment Statistical Report

The stability of PIPE's forecasting performance is validated through standard deviation analysis across three random seeds, reported in Tables Table 7 and Table 8.

## E.3 Scaling Behavior

To evaluate the impact of model size on performance, we conduct experiments across three variants: PIPE-3B, PIPE-7B, and PIPE-32B (with LoRA rank as 8). As demonstrated in Tables 9 and 10, the largest model, PIPE-32B, yields performance improvements, even when leveraging LoRA.

Table 7: The mean and the standard deviation of PIPE-3B from three random seeds (leading time is 6 hours).

| Models | Intensity (hPa) | | Latitude (°) | | Longitude (°) | | Distance (km) |
|---|---|---|---|---|---|---|---|
| | MAE | RMSE | MAE | RMSE | MAE | RMSE | MAE |
| PIPE-3B-seed1 | 1.503 | 2.940 | 0.085 | 0.161 | 0.095 | 0.178 | 16.364 |
| PIPE-3B-seed2 | 1.513 | 2.946 | 0.082 | 0.154 | 0.094 | 0.173 | 16.000 |
| PIPE-3B-seed3 | 1.529 | 3.059 | 0.084 | 0.161 | 0.097 | 0.181 | 16.463 |
| PIPE-3B | $1.515 \pm 0.011$ | $2.981 \pm 0.055$ | $0.084 \pm 0.001$ | $0.159 \pm 0.003$ | $0.095 \pm 0.001$ | $0.178 \pm 0.003$ | $16.275 \pm 0.200$ |

Table 8: The mean and the standard deviation of PIPE-3B from three random seeds (leading time is 12 hours).

| Models | Intensity (hPa) | | Latitude (°) | | Longitude (°) | | Distance (km) |
|---|---|---|---|---|---|---|---|
| | MAE | RMSE | MAE | RMSE | MAE | RMSE | MAE |
| PIPE-3B-seed1 | 3.840 | 6.295 | 0.362 | 0.530 | 0.412 | 0.590 | 67.432 |
| PIPE-3B-seed2 | 3.831 | 6.281 | 0.355 | 0.523 | 0.405 | 0.578 | 66.402 |
| PIPE-3B-seed3 | 3.893 | 6.425 | 0.360 | 0.527 | 0.415 | 0.592 | 67.506 |
| PIPE-3B | $3.855 \pm 0.027$ | $6.333 \pm 0.065$ | $0.359 \pm 0.003$ | $0.526 \pm 0.003$ | $0.411 \pm 0.004$ | $0.587 \pm 0.006$ | $67.114 \pm 0.050$ |

Table 9: The results of PIPE-3B, PIPE-7B, and PIPE-32B (with LoRA) with the lead time of 6 hours.

| Models | Intensity (hPa) | | Latitude (°) | | Longitude (°) | | Distance (km) |
|---|---|---|---|---|---|---|---|
| | MAE | RMSE | MAE | RMSE | MAE | RMSE | MAE |
| PIPE-3B | 1.515 | 2.981 | 0.084 | 0.159 | 0.095 | 0.178 | 16.275 |
| PIPE-7B | 1.505 | 2.918 | 0.088 | 0.166 | 0.102 | 0.184 | 17.194 |
| PIPE-32B | 1.505 | 2.874 | 0.079 | 0.153 | 0.097 | 0.182 | 15.980 |

### E.4 Forecasting Results on Different Grades

We experiment to investigate the performance of our method on different grades of typhoons. We split the dataset based on the grade (from grade 2 to grade 6). Grade 2 represents tropical depressions, which are weaker cyclones and not classified as tropical cyclones. Grades 3, 4, and 5 represent tropical cyclones, with Grade 5 being the most intense. Grade 6 represents a cyclone with a different structural system from tropical cyclones. Then we train models on different sub-datasets. The results are shown in Table 11.

**Track Forecasting vs. Pressure Forecasting**    Tropical cyclones (Grades 3, 4, 5) demonstrate better track forecasting performance compared to non-tropical cyclones (Grades 2, 6). This is likely due to the more stable and predictable system of tropical cyclones, which facilitates track forecasting. However, pressure forecasting is more challenging for tropical cyclones due to higher variability and complexity. In contrast, non-tropical cyclones (Grade 2 and Grade 6) exhibit more predictable pressure patterns but greater difficulty in track forecasting. This is especially pronounced before the cyclone structure forms (Grade 2) and after it breaks (Grade 6).

**Impact of Intensity on Forecasting**    As the intensity of tropical cyclones increases (Grades 3, 4, 5), pressure forecasting becomes progressively more challenging. However, the difficulty of track forecasting remains relatively consistent across these grades.

**Impact of Data Volume**    Models trained on all grades perform better in track forecasting compared to models trained on individual grades. This result indicates that a larger volume of training data improves track forecasting accuracy across different cyclone intensities.

### E.5 Experiment on the Offset Indexing

In addition to negative indexing, we investigate an alternative approach called offset indexing, where all image tokens are assigned an offset equal to the maximum length of the textual tokens to address conflicts between the physical information of vision tokens and the order information of text tokens.

Table 10: The results of PIPE-3B, PIPE-7B, and PIPE-32B (with LoRA) with the leading time of 12 hours.

| Models | Intensity (hPa) | | Latitude (°) | | Longitude (°) | | Distance (km) |
|--------|------|------|------|------|------|------|------|
| | MAE | RMSE | MAE | RMSE | MAE | RMSE | MAE |
| PIPE-3B | 3.855 | 6.333 | 0.359 | 0.526 | 0.411 | 0.587 | 67.114 |
| PIPE-7B | 3.861 | 6.325 | 0.371 | 0.540 | 0.435 | 0.609 | 69.933 |
| PIPE-32B | 3.695 | 6.029 | 0.342 | 0.510 | 0.423 | 0.610 | 66.725 |

Table 11: The results of the experiment across different grades with the lead time of 6 hours.

| Models | Intensity (hPa) | | Latitude (°) | | Longitude (°) | | Distance (km) |
|--------|------|------|------|------|------|------|------|
| | MAE | RMSE | MAE | RMSE | MAE | RMSE | MAE |
| Grade 2 | 0.722 | 1.227 | 0.108 | 0.206 | 0.138 | 0.245 | 22.029 |
| Grade 3 | 0.997 | 1.772 | 0.097 | 0.169 | 0.115 | 0.200 | 18.902 |
| Grade 4 | 1.656 | 2.732 | 0.099 | 0.156 | 0.122 | 0.188 | 19.310 |
| Grade 5 | 2.707 | 4.556 | 0.090 | 0.163 | 0.116 | 0.203 | 18.418 |
| Grade 6 | 1.074 | 1.830 | 0.208 | 0.385 | 0.283 | 0.488 | 37.193 |

The results of this experiment are presented in Table 12. We find that although offset indexing offers some performance improvement, it falls short compared to PIPE. Our choice of negative indexing remains an intuitive and efficient method for effectively separating different types of tokens.

### E.6 Showcase

We present a prediction showcase (Figure 6) to compare our method with the methods that remove satellite imagery and PIPE on the Typhoon Phanfone. PIPE achieves more accurate track forecasting and intensity forecasting.

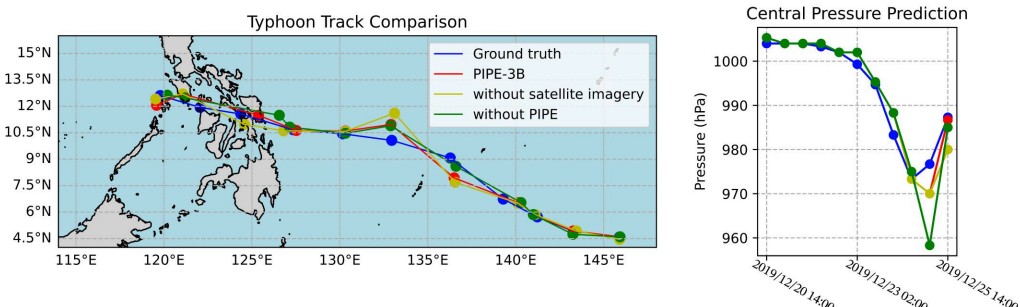

Figure 6: The results of Typhoon Phanfone comparison between PIPE, removing satellite images, and removing PIPE. The leading time is 12 hours and the time gap between neighbouring dots is 12 hours.

### E.7 Attention Analysis

We compare the attention from the penultimate layer (Figure 7) with averaging across the head dimension of PIPE-3B and Qwen2.5-VL-3B. It reveals distinct attention patterns. Qwen2.5-VL-3B exhibits an obvious bias toward the initial tokens of the historical time series, as evidenced by an obvious vertical line at 800th input tokens ((e) & (f)). In contrast, our PIPE model allocates greater attention to both the image tokens and the historical time series tokens. Notably, PIPE's attention on image patches is concentrated on the typhoon region (e.g., central cloud structure), whereas

Table 12: The results of offset indexing.

| Models | Intensity (hPa) | | Latitude (°) | | Longitude (°) | | Distance (km) |
|---|---|---|---|---|---|---|---|
| | MAE | RMSE | MAE | RMSE | MAE | RMSE | MAE |
| no-indexing | 1.961 | 3.926 | 0.206 | 0.360 | 0.388 | 0.674 | 53.548 |
| offset indexing | 1.815 | 3.762 | 0.188 | 0.303 | 0.229 | 0.385 | 30.157 |
| negative indexing (PIPE) | 1.515 | 2.981 | 0.084 | 0.159 | 0.095 | 0.178 | 16.275 |

Qwen2.5-VL-3B's attention appears diffuse and unstructured across the image. These differences in attention mechanisms likely contribute to PIPE's better forecasting accuracy.

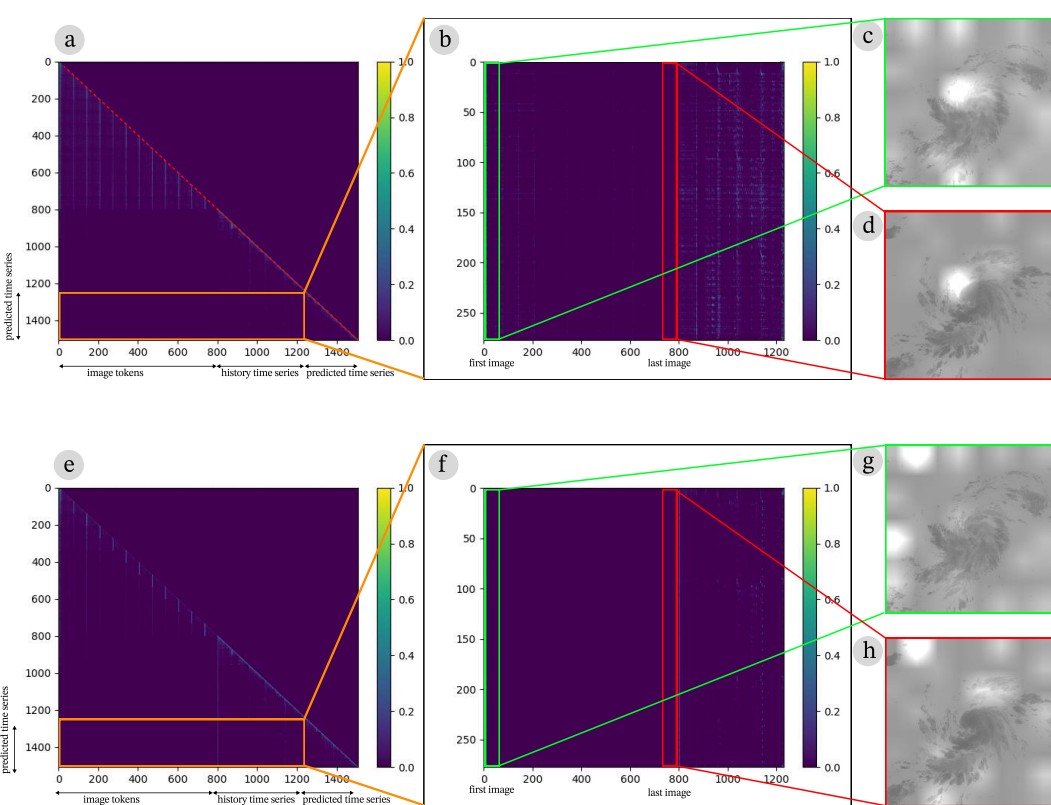

Figure 7: Visualization of attention (normalized to 0-1 in each step) from the penultimate layer of the PIPE model (top) and the Qwen2.5-VL-3B model (bottom), averaged across attention heads. (a) & (e) The entire attention matrix. (b) & (f) The attention matrix of predicted tokens' attention on the input tokens, including image tokens and history time series tokens. (c) & (g) Attention of predicted tokens on the first input image. (d) & (h) Attention of predicted tokens on the last input image.

We also compare the attention using Attention Rollout [1] (Figure 8) with averaging across the head dimension of PIPE-3B and Qwen2.5-VL-3B. It also demonstrates that our model allocates more reasonable attention to image tokens and historical time series tokens. Furthermore, our model's attention on image patches is focused specifically on the typhoon region, whereas Qwen2.5-VL-3B's attention appears biased across the image.

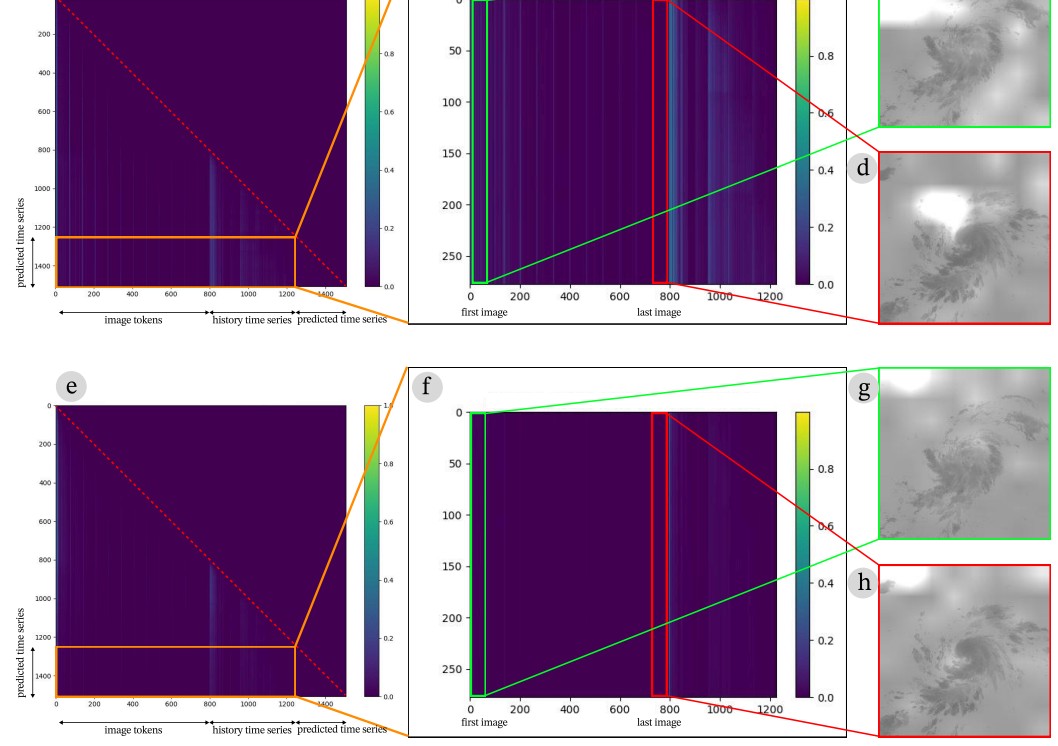

Figure 8: Visualization of attention (normalized to 0-1 in each step) using Attention Rollout of the PIPE model (top) and the Qwen2.5-VL-3B model (bottom), averaged across attention heads. (a) & (e) The entire attention matrix. (b) & (f) The attention matrix of predicted tokens' attention on the input tokens, including image tokens and history time series tokens. (c) & (g) Attention of predicted tokens on the first input image. (d) & (h) Attention of predicted tokens on the last input image.

# F  Broader Impact

We introduce a novel multimodal time series forecasting task that integrates satellite imagery with temporal data for capturing complex spatio-temporal dependencies. This approach leverages the complementary strengths of temporal time series data and spatially rich visual inputs, enabling models to go beyond the limitations of traditional univariate, multivariate, or single-modality methods. To address the inherent challenges of integrating satellite imagery into time series forecasting, we propose a physics-informed positional encoding. This technique incorporates physical information derived from satellite data, such as geospatial coordinates, to enhance the model's ability to reason about spatial and temporal dependencies. This innovation is particularly relevant for applications where visual inputs carry critical physical context, including climate modeling, urban planning, and agricultural forecasting. The broader impact of this work lies in its ability to bridge the gap between traditional forecasting methods and real-world complexities that often include spatial and physical components. By incorporating satellite imagery and physics-informed encoding, this method has potential benefits across a wide range of scientific and practical domains.

