# OpenReview forum: "PIPE: Physics-Informed Position Encoding for Alignment of Satellite Images and Time Series in Typhoon Forecasting"
_NeurIPS.cc/2025/Conference — NeurIPS 2025 poster_

### Official Review · Reviewer_fqdc · 2025-06-25

**Clarity:** 2
**Significance:** 3
**Originality:** 3
**Rating:** 4
**Confidence:** 3

**Summary:**

They propose a method to align different modalities which is physics informed, taking into account spatial and temporal information. Additionally, a separate positional encoding that uses variant-frequency is introduced for the physical variables. These should ideally help in typhoon intensity forecasting with the ability to integrate satellite imagery, which was not yet done before. The paper shows experiments that illustrate improvement over existing models, including models currently being used for current weather forecasting models. Ablations also show the contribution of each component of the proposed model.

**Questions:**

- why use Qwen-2.5-vl for the text and vision encoders?
- what’s the motivation for freezing the vision encoder and not the text encoder?
- on temporal position IDs: time has a “wrap around” property where the start of the next year is close to the end of the previous year. How is this taken into account (Eq 7)? Related, why are different components using different types of position IDs—why not just use the same type for all inputs?
- Line 262-263: what is meant by the original dataset publication? is it the Digital typhoon dataset with the typhoon track annotations from Best Track?
- Is the Qwen-2.5-VL baseline trained the same way as PIPE but without the multimodal alignment enhancements?
- The PIPE contribution/improvement on the latitude and longitudes don’t seem big perhaps using UTM would better quantify the improvement (Tables 1,2)? 1 degree of lat/lon are quite large so the differences could be better emphasized in UTM since it’s in meters

**Ethical Concerns:**

["Major Concern: Data quality and representativeness"]

**Final Justification:**

The paper introduces a new positional encoding method that takes into account the physics. I believe the paper supports this contribution sufficiently and shows comprehensive experiments and ablations to show the benefit of their method. The authors' rebuttal also addressed my concerns and most of the other reviewers' concerns, so I would like to maintain my positive review.

**Limitations:**

The paper mostly talks about results having 12h lead time, is this generally enough to warn relevant agencies and for decision-making?
The distribution of the data was also not discussed which could pose limitations on the types of locations that this model can be applicable to.

**Quality:**

3

**Strengths And Weaknesses:**

Strengths
- Problem is well-motivated
- Improvement with the addition of satellite image in forecasting
- Prediction of typhoon positions seem accurate (Fig 3)
- Ablation study shows contribution of each component of the proposed model
- Extensive experiments that analyze the different aspects of PIPE such as showing decreasing errors with decreasing leading times, showing model improves with more introduced data (Appendix, Fig 5)

Weaknesses
- Design choices need to be further explained — i.e., instead of just saying “x was used…”, the rationale behind the choice could be further details (e.g., using Qwen as the text and vision encoders, freezing vision encoder, not freezing text encoder, using different positional indexing depending on the type of data such as sequential, 3D indexing, physics informed indexing, variant-frequency positional encoding)
- Implementation details are not very clear, especially on the Qwen baseline and how it was trained, and how exactly it differs from PIPE —- is it just the position encoding that’s different?

---

> ### Author Rebuttal · Authors · 2025-07-31
>
> Dear reviewer [fqdc],
>
> Thank you for your efforts and the positive feedback.
>
> > Weakness #1: Design choices need to be further explained — i.e., instead of just saying “x was used…”, the rationale behind the choice could be further details (e.g., using Qwen as the text and vision encoders, freezing vision encoder, not freezing text encoder, using different positional indexing depending on the type of data such as sequential, 3D indexing, physics informed indexing, variant-frequency positional encoding)
> >
> > Question #1: why use Qwen-2.5-vl for the text and vision encoders?
> >
> > Question #2: what’s the motivation for freezing the vision encoder and not the text encoder?
>
> Thank you for your advice. We will include more detailed explanations of the selection of Qwen and freezing the vision encoder in the revised paper:
> 1. Why use Qwen:
>     * This model incorporates 3D positional encoding (3D PE), which enhances its ability to align with video understanding. This feature is particularly relevant to our satellite image sequences that require such capabilities, making Qwen a suitable choice.
>     * Qwen demonstrates strong performance in multimodal alignment, which is crucial for our work, such as the MMMU benchmark [1].
>     * It is an open-source model.
> 2. Motivation for freezing the vision encoder and not the text encoder:
>     * We need to fine-tune the LLM during the SFT stage to equip it with domain-specific knowledge, ensuring it becomes trainable for our tasks.
>     * For the vision encoder, we observed that the original encoder is already effective in capturing visual patterns, and additional training did not yield further benefits. The experimental results are as follows:
> |Model|Intensity MAE|Intensity RMSE|Latitude MAE|Latitude RMSE|Longitude MAE|Longitude RMSE|Distance MAE|
> |---|---|---|---|---|---|---|---|
> |unfreezing the vision encoder|1.633|3.307|0.084|0.155|0.101|0.184|16.696|
> |PIPE (freezing)|1.515|2.981|0.084|0.159|0.095|0.178|16.275|
>
> 3. Regarding the selection of different components, we have conducted a comprehensive ablation study to validate their effectiveness, as highlighted in your **Strength #4**. The results demonstrate that each selection of the components is reasonable.
>
> > Weakness #1: Implementation details are not very clear, especially on the Qwen baseline and how it was trained, and how exactly it differs from PIPE —- is it just the position encoding that’s different?
> >
> > Question #5: Is the Qwen-2.5-VL baseline trained the same way as PIPE but without the multimodal alignment enhancements?
>
> Thank you for your feedback. Yes, Qwen and PIPE differ only in their positional encoding to evaluate the effects of our methods. Both were implemented using LLama-Factory on NVIDIA H800 GPUs, optimized with AdamW, an initial learning rate of 1e-5, a batch size of 1, and cross-entropy loss over one training epoch. **The code has been included in our submission.**
>
> These details are also outlined in Section 4.2 (line 264) and Appendix C: Implementation Details.
>
> > Question #3: on temporal position IDs: time has a “wrap around” property where the start of the next year is close to the end of the previous year. How is this taken into account (Eq 7)? Related, why are different components using different types of position IDs—why not just use the same type for all inputs?
>
> Thank you for your question.
> 1. In the Eq7, t is the hourly progression of the year.  During training, the attention mechanism can capture the relationship between the end of one year and the start of the next if there are "wrap-around" patterns present in the training dataset.
> 2. We used different types of positional IDs for textual tokens and vision tokens. Vision tokens encode satellite image patches that contain physical information such as time, latitude, and longitude, while textual tokens only encode their sequential order.
>
> > Question #4: Line 262-263: what is meant by the original dataset publication? is it the Digital typhoon dataset with the typhoon track annotations from Best Track?
>
> Thank you for your question.
> 1. The "original dataset publication" refers to the open-sourced dataset cited in [2].
> 2. Yes, the annotations in this dataset are derived from the Best Track dataset, as described in Section 4.1 (line 244).
>
> > Question #5: The PIPE contribution/improvement on the latitude and longitudes don’t seem big perhaps using UTM would better quantify the improvement (Tables 1,2)? 1 degree of lat/lon are quite large so the differences could be better emphasized in UTM since it’s in meters
>
> Thank you for your suggestion.
> 1. Using longitude and latitude is a standard practice in typhoon research, especially given the large scale of the study (as illustrated in Figure 1).
> 2. Moreover, the data were normalized in the model, which allows the **differences to be effectively emphasized**.
> 3. We agree that converting coordinates to UTM may offer further advantages. At this stage, meanwhile, we believe that the choice of coordinate transformation does not materially affect the core contribution of our work, as our focus is on validating the effectiveness of the physics-informed position encoding framework itself. We will exploring alternative coordinate systems, include UTM, in our future work.
>
> > Limitation #1: The paper mostly talks about results having 12h lead time, is this generally enough to warn relevant agencies and for decision-making? The distribution of the data was also not discussed which could pose limitations on the types of locations that this model can be applicable to.
> >
> > Ethical Concerns: Data quality and representativeness
>
> Thank you for your question.
> 1. **Short-term forecasting is of critical importance** for timely decision-making, as typhoon events typically span only a few days. For example, the maximum forecasting length is limited to 12 hours in the dataset baseline [2] used in our study and limited to 1 hour in [3], which aligns with the practical need for short-term predictions.
>
>     We have also included the extended forecasting horizons for VLMs in the Limitations section (line 642). Addressing the challenge of long-term forecasting will be one of the key focuses of future work, particularly addressing the limitation of current VLMs to handle more input images. However, the **primary goal of this paper is to explore whether PIPE is effective** in incorporating additional physical information into forecasting models. We believe that our experimental results successfully demonstrate the validity of our methods.
> 2. The quality of the dataset: The dataset [2] we used is of high quality, as it was recognized with a Spotlight at NIPS. Additionally, the dataset is representative, encompassing all historical typhoons from 1978 to 2023.
>
> \
> Finally, we want to thank you once again for your thoughtful comments. We look forward to your feedback and we kindly ask that you consider these clarifications in your final grading.
>
> Warm regards,
>
> The Authors
>
> \
> [1] Yue, Xiang, et al. "Mmmu: A massive multi-discipline multimodal understanding and reasoning benchmark for expert agi." Proceedings of the IEEE/CVF Conference on Computer Vision and Pattern Recognition. 2024.
>
> [2] Kitamoto, Asanobu, et al. "Digital typhoon: Long-term satellite image dataset for the spatio-temporal modeling of tropical cyclones." Advances in Neural Information Processing Systems 36 (2023): 40623-40636.
>
> [3] Zhu, Jiakai, and Jianhua Dai. "A rain-type adaptive optical flow method and its application in tropical cyclone rainfall nowcasting." Frontiers of Earth Science 16.2 (2022): 248-264.

---

> > ### Author Response · Authors · 2025-08-05
> >
> > Dear Reviwer [fqdc],
> >
> > Thank you once again for your positive feedback. To address your concerns regarding design choices, implementation details, and the need for further clarification, we have conducted an additional experiment and provided more detailed explanations.
> >
> > As the discussion deadline approaches, please feel free to reach out with any further questions or if additional clarification is needed. We would be more than happy to engage in further discussion. If you find that we have addressed your concerns, we kindly request your consideration in improving our score. Thank you very much.
> >
> > \
> > Best regards,
> >
> > Authors

---

> > > ### Comment · Reviewer_fqdc · 2025-08-06
> > >
> > > Thank you for the clarification. The rebuttal has addressed my concerns, and I have no further questions. The additional experiments on other applications such as AU Typhoon and crop monitoring are helpful to support the generalizability/effectivity of the method, and it would be great to include these in the final version of the paper.

---

> ### Author Response · Authors · 2025-08-06
>
> Dear Reviewer [fqdc],
>
> Thank you for your thoughtful feedback and for acknowledging that **we have addressed all your concerns**. We will ensure that **these revisions are incorporated into the final version** of the paper. We would be **truly grateful if you could consider increasing your recommendation to a higher level**, as your feedback has been valuable in enhancing the quality of our work.
>
> Thank you once again for your valuable comments and suggestions.
>
> \
> Best regards,
>
> Authors

---

### Official Review · Reviewer_iSKv · 2025-06-30

**Clarity:** 2
**Significance:** 2
**Originality:** 2
**Rating:** 4
**Confidence:** 3

**Summary:**

This paper proposes PIPE (Physics-Informed Position Encoding), a method that improves multimodal time series forecasting by embedding physical metadata—such as time, latitude, and longitude—into a vision-language model. PIPE consists of two key components: a physics-informed positional indexing scheme and a variant-frequency sinusoidal encoding that reflect real-world spatiotemporal structures. Applied to typhoon forecasting using satellite images and numerical data, PIPE outperforms existing models, achieving up to 12% improvement in intensity prediction.

**Questions:**

This reviewer would like to recommend that the authors please:

1. Clarify how PIPE is novel compared to existing encoding methods like RoPE, relative position encodings, and 3D indexing.
1. Provide evidence (e.g., ablations or probing) that the model meaningfully uses the injected physical encodings.
1. Include experiments on PIPE’s generalizability to domains beyond typhoon forecasting.
1. Compare PIPE to standard positional encoding baselines such as learned embeddings and RoPE.

**Ethical Concerns:**

["NO or VERY MINOR ethics concerns only"]

**Final Justification:**

While the paper formulates an interesting multimodal time series forecasting task and the proposed methods (PIPE) are easy to implement, the work as it presently stands has several weaknesses that my review mentions. Although authors’ rebuttal tried to address some of them, I remain unconvinced that there is a fundamentally new mechanism here. The proposed position encoding approach reuses well-established principles without introducing any fundamentally new mechanisms, and the authors rebuttal reinforces that assessment. The follow up response by the authors essentially reiterate the same points from the original rebuttal without providing new insights and thus does not not alleviate my fundamental concern.

However, after looking at the exchange between the authors and the other reviewers as well as the clear message from the PC Chairs about seeking convergence, I have decided to suspend my concerns and have improved the rating from Borderline Reject to Borderline Accept, which is in line with other reviewers.

**Limitations:**

yes

**Quality:**

2

**Strengths And Weaknesses:**

**Strengths:**

1. The paper clearly formulates a novel multimodal time series forecasting task that integrates satellite imagery with numerical sequences, addressing a realistic and impactful use case in climate forecasting.
1.  PIPE is easy to implement and requires no additional learnable parameters, making it broadly applicable to existing vision-language models with minimal overhead.

**Weaknesses:**
1. While PIPE delivers promising empirical gains in multimodal time series forecasting, its core methodological design offers limited novelty. The physics-informed positional indexing maps known physical variables—such as timestamps, latitude, and longitude—to token position IDs using deterministic rules. This approach is structurally similar to absolute and 3D positional encoding schemes previously used in Transformers and Vision Transformers. Likewise, the variant-frequency sinusoidal encoding extends the standard Transformer positional encoding by introducing customized frequencies for each physical variable (e.g., hourly or geographic periodicity). While this results in a tailored encoding of spatiotemporal information, it remains an intuitive modification of the classic sinusoidal formulation.
1. In addition, because these encodings are non-learnable and differ substantially from the token patterns observed during pretraining, it is unclear how effectively the underlying vision-language model can interpret them. Pretrained models like Qwen-2.5-VL are unlikely to have encountered structured positional patterns grounded in real-world physics (e.g., hourly timestamps mapped to unique negative IDs), and no mechanisms are introduced to ensure the model learns to associate these encodings with physical meaning. The integration thus relies entirely on the model’s ability to infer these semantics during fine-tuning, which may be unreliable. Without interpretability studies showing the model's reliance on the injected encodings, there is a risk that these handcrafted inputs are underutilized.
1. The evaluation in this paper is limited to a single domain—typhoon forecasting using the Digital Typhoon dataset—which raises concerns about the generalizability of PIPE. Although the dataset is large and well-annotated, it represents only one type of spatiotemporal phenomenon under relatively consistent geographic and temporal conditions. It remains unclear whether PIPE can adapt to other multimodal forecasting tasks that differ in dynamics, data quality, or modality structure, such as precipitation estimation, air quality prediction, urban heat mapping, or traffic forecasting. These applications may involve different temporal resolutions, irregular sampling, heterogeneous sensor data, or weaker correlations between image and numerical modalities.
1. The paper lacks a thorough comparison to alternative and widely-used positional encoding strategies, which limits the strength of its empirical claims. While ablation studies demonstrate the contribution of PIPE’s components (e.g., physics-informed indexing and variant-frequency encoding), the evaluation does not include baselines using other strong positional encoding methods, such as learned positional embeddings, relative positional encodings, or Rotary Positional Encoding (RoPE), which has proven effective in both language and vision tasks. These methods are well-established, often adaptable, and have demonstrated strong performance in capturing sequence order or spatial structure, including in long-context or multimodal settings. Moreover, since PIPE is non-learnable and fixed, while others are adaptive or trainable, it would be particularly valuable to compare performance, robustness, and interpretability across these approaches.

---

> ### Author Rebuttal · Authors · 2025-07-31
>
> Dear Reviewer [iSKv],
>
> Thank you for your time and comments.
>
> > Weakness #1: While PIPE delivers promising empirical gains in multimodal time series forecasting, its core methodological design offers limited novelty. ... it remains an intuitive modification of the classic sinusoidal formulation.
> >
> > Question #1:Clarify how PIPE is novel compared to existing encoding methods like RoPE, relative position encodings, and 3D indexing.
>
> Thank you for acknowledging the empirical gains achieved by our method and for providing feedback on its novelty. The novelty of our method lies in its incorporation of the **global physical information** for vision tokens.
>
> 1. **Comparison**:
>     * Existing positional encoding methods, including RoPE [1], relative PEs [2, 3] and 3D indexing [4], primarily focus on representing the absolute or relative order of tokens **within a single input sample**. These include absolute and relative positional encoding schemes, all of which operate at the level of individual samples and are not designed to encode global information. In contrast, our method captures **global physical information shared across all input samples**. It enables the model to better capture spatiotemporal relationships and leverage domain knowledge effectively among all input samples, as the physical knowledge remains unchanged.
>
>     * Additionally, we designed the encoding to encode **characteristics of specific physical variables**, allowing it to capture domain-specific periodicities (e.g., hourly or geographic cycles). This capability is limited or absent in existing positional encoding schemes and represents another key distinction of our approach.
>     * We have provided these clarifications in the **Introduction** (the paragraph prior to the contributions) and the **Related Work** section (‘Position Encoding in Transformers’). We appreciate the opportunity to clarify these points.
>
> 2. **Novelty**: We respectfully argue that the value and novelty of a method should not be judged solely by whether it appears intuitive. Intuitive methods can be highly impactful, particularly when they result in substantial performance improvements, as demonstrated by our results. We believe that is the reason that Reviewer [8J4Q] and Reviewer [hPaZ] think we have a novel and elegant design of the scheme.
>
> > Weakness #2: In addition, because these encodings are non-learnable and differ substantially from the token patterns observed during pretraining, it is unclear how effectively the underlying vision-language model can interpret them. ... Without interpretability studies showing the model's reliance on the injected encodings, there is a risk that these handcrafted inputs are underutilized.
> >
> >  Question #2: Provide evidence (e.g., ablations or probing) that the model meaningfully uses the injected physical encodings.
>
> Thank you for your feedback. We have used different ways to provide the evidence (including the ablations you suggested, and the attention visualization) to prove that our model meaningfully uses the physical informed encodings.
>
> 1. Ablation Study:
> In our ablation study, we systematically remove components one by one, such as 3D indexing, physics-informed indexing, and the variant-frequency sinusoidal function, to demonstrate they are meaningfully used. Reviewer [fqdc] also commented that our ablation study is comprehensive.
>
> 2. Attention Visualization (Section E.5) for Interpretability:
> We visualized the attention scores to further explain the model's behavior. Our analysis shows that with PIPE, the model's attention is concentrated on the typhoon region, whereas without PIPE, the attention becomes more diffuse. It demonstrates that the visual structure of typhoons are used for forecasting with PIPE.
>
> These analyses provide evidence that the physical encodings meaningfully contribute to the model's performance.
>
> > Weakness #3: The evaluation in this paper is limited to a single domain ... These applications may involve different temporal resolutions, irregular sampling, heterogeneous sensor data, or weaker correlations between image and numerical modalities.
> >
> > Question #3: Include experiments on PIPE’s generalizability to domains beyond typhoon forecasting.
>
> Thank you for your suggestion. To demonstrate the generalizability of our method, we added two additional experiments targeting different application domains: one focuses on typhoon forecasting in another region (the Australian (AU) region) [5]; the other uses the T31TFM-16 dataset for crop monitoring [6]. The first experiment confirms the robustness of our method in typhoon forecasting, while the second demonstrates its applicability to a different domain, showcasing its broader generalization capability.
>
> 1. AU typhoon experiment:
>   * Settings: The settings remain the same as PIPE. The focus is on typhoon forecasting, a regression task, which is the core of our paper.
>   * Result:
> |Models|Intensity MAE|Intensity RMSE|Latitude MAE|Latitude RMSE|Longitude MAE|Longitude RMSE|Distance MAE|
> |---|---|---|---|---|---|---|---|
> |Qwen-2.5-VL-3B (after tuning)|1.399|2.806|0.098|0.201|0.121|0.211|19.718|
> |PIPE-3B (Zero-shot)|1.361|2.773|0.094|0.171|0.118|0.206|19.090|
> |PIPE-3B (after tuning)|1.352|2.558|0.093|0.163|0.116|0.202|18.835|
>
>   * Analysis:
> The results show that our method can be generalized to different regions for typhoon forecasting. Even the **model trained on the West Pacific region can perform well in the new dataset (zero-shot)**. For instance, compared to Qwen-2.5-VL-3B (after tuning), PIPE-3B (Zero-shot) achieved a lower intensity RMSE (2.773 vs. 2.806) and a smaller distance error (19.090 vs. 19.718). Additionally, PIPE-3B (after tuning) outperformed both models, achieving the best overall performance.
>
> 2. Crop monitoring experiment:
>   * Setting: We adopt our method to the semantic segmentation task. The training data and test data were split by the original dataset, with a ratio of 0.85:0.15. We calculate latitude and longitude from the given UTM information provided in the dataset. We use the ViT model to encode the satellite image patches and incorporate each patch’s physical information. The focus is on crop monitoring, a segmentation task. Number of epochs: 50. lr: 1e-4. Optimizer: Adam.
>
>   * Result:
> |Models|IoU|Accuracy|
> |---|---|---|
> |ViT PE|67.52|80.61|
> |3D PE|67.27|80.43|
> |PIPE|69.21|81.81|
>
>   * Analysis: The results show that our method achieves the best result (69.21% and 81.81%) than standard ViT PE (67.52% and 80.61%)  and 3D PE (67.27% and 80.43%). The experiment shows that our method can be generalized to other domains by incorporating global physical information.
>
> > Weakness #4: The paper lacks a thorough comparison to alternative and widely-used positional encoding strategies, which limits the strength of its empirical claims.  ... it would be particularly valuable to compare performance, robustness, and interpretability across these approaches.
> >
> > Question #4: Compare PIPE to standard positional encoding baselines such as learned embeddings and RoPE.
>
> We appreciate the reviewer's insightful feedback regarding positional encoding comparisons.
> 1. We have compared our method with RoPE in the paper. In the main experiment (Table 1), ‘Qwen-2.5-VL-3B’ and in the ablation study (Table 2), ‘w/o 3D indexing (using sequence)’ are both using RoPE. We will add the clarification in the revised paper.
>
> 2. Regarding the use of learned PE, we evaluated it prior to designing our main experiments.
>   * As noted in the literature [4, 7],  learnable PEs do not outperform deterministic ones. For instance, [7] states: “We also experimented with using learned positional embeddings instead and found that the two versions produced nearly identical results.”
>   * Another reason for not using learnable PEs in VLMs is that the **input sequence length can vary and may exceed** the maximum sequence length encountered during training. In such cases, learned PEs may fail to generalize.
>
>     These findings are reflected in modern architectures:
>     * RoPE: Used in Qwen [4] and LLaMA [8].
>     * Sinusoidal: Original Transformer [7] and other variants.
>
>     Therefore, we opted for deterministic PEs, as they offer the potential for the model to extrapolate to sequence lengths beyond those seen during training.
>
> \
> Finally, we want to thank you once again for your thoughtful comments. We look forward to your feedback and we kindly ask that you consider these clarifications in your final grading.
>
> Warm regards,
>
> The Authors
>
> \
> [1] Su, Jianlin, et al. "Roformer: Enhanced transformer with rotary position embedding." Neurocomputing 568 (2024): 127063.
>
> [2] Shaw, Peter, Jakob Uszkoreit, and Ashish Vaswani. "Self-attention with relative position representations." arXiv preprint arXiv:1803.02155 (2018).
>
> [3] Ke, Guolin, Di He, and Tie-Yan Liu. "Rethinking positional encoding in language pre-training." arXiv preprint arXiv:2006.15595 (2020).
>
> [4] Bai, Shuai, et al. "Qwen2. 5-vl technical report." arXiv preprint arXiv:2502.13923 (2025).
>
> [5] Kitamoto, Asanobu, Erwan Dzik, and Gaspar Faure. "Machine Learning for the Digital Typhoon Dataset: Extensions to Multiple Basins and New Developments in Representations and Tasks." arXiv preprint arXiv:2411.16421 (2024).
>
> [6] Tarasiou, Michail, Riza Alp Güler, and Stefanos Zafeiriou. "Context-self contrastive pretraining for crop type semantic segmentation." IEEE Transactions on Geoscience and Remote Sensing 60 (2022): 1-17.
>
> [7] Vaswani, Ashish, et al. "Attention is all you need." Advances in neural information processing systems 30 (2017).
>
> [8] Grattafiori, Aaron, et al. "The llama 3 herd of models." arXiv preprint arXiv:2407.21783 (2024).

---

> > ### Comment · Reviewer_iSKv · 2025-08-05
> >
> > I appreciate at the authors' effort at responding to my review. Unfortunately, I do not find the rebuttal convincing, mainly due to the following two points. Therefore I will unfortuantely not be able to review the score upwards.
> >
> > **Novelty Still Not Convincing:** Although the authors highlight PIPE’s use of global physical variables such as time, latitude, and longitude for position encoding, the core design largely builds on established methods. The physics-informed indexing is structurally similar to 3D indexing schemes that assign positional IDs along temporal and spatial axes, while the variant-frequency sinusoidal encoding is a straightforward adaptation of standard sinusoidal position encodings with variable periodicities. These components reuse well-known positional encoding principles without introducing fundamentally new mechanisms. Therefore, the contribution is better characterized as an application-specific adaptation rather than a novel encoding method.
> >
> > **Comparison to Standard Positional Encoding Baselines:** While the authors claim that PIPE is compared to RoPE and that learned positional embeddings typically underperform, these comparisons are neither clearly presented nor systematically evaluated. RoPE appears to be used implicitly in certain ablation settings (e.g., sequence-based indexing), but there is no dedicated, controlled experiment that directly contrasts PIPE with RoPE or learned embeddings under otherwise identical conditions. Without explicit baselines and side-by-side comparisons, it is difficult to assess the actual benefit of PIPE over standard alternatives. This lack of rigorous benchmarking limits the strength of the empirical evidence supporting PIPE’s effectiveness.

---

> > > ### Author Response · Authors · 2025-08-05
> > >
> > > Dear Reviewer [iSKv],
> > >
> > > Thank you for your feedback. We are pleased that two concerns (generalizability and interpretability) are addressed by our additional experiments and clarifications. We also appreciate your further response regarding the novelty and the comparisons to RoPE. Below, we aim to address with further clarification and hope this will fully address your remaining concerns.
> > >
> > > 1. Novelty:
> > >
> > >     * We appreciate your acknowledgment of our use of additional global physical information for PE. As you noted, PIPE builds on existing LLM designs; however, we believe our contribution lies in the incorporation of new physical information and the novel design for its integration. This novelty of the design has also been positively highlighted by other reviewers (e.g., Reviewers [8J4Q] and [hPaZ]).
> > >
> > >     * We **understand your perspective and will revise the manuscript to better characterize PIPE** as an application-specific adaptation, as you have suggested.
> > >
> > > 2. Comparison:
> > >
> > >     * We had a comprehensive ablation study (Reviewer [fqdc]) in our paper, including RoPE. Your concern about RoPE being used in certain ablation settings is decided by itself, as **position indexing is a part of RoPE**. The core of RoPE is to rotate the inputs according to the position indexing. However, removing the position indexing while implementing RoPE is inherently challenging, as **the position indexing is fundamental to RoPE’s operation**. **RoPE cannot function as designed without position indexing**, as the positional indices are required to compute the relative rotations that encode positional relationships. For instance, the original RoPE paper [1] used sequence indexing, while the Qwen technical report [2] adopted 3D indexing.
> > >
> > >     * For learned positional embeddings, we provide a rationale for their limited use in current LLMs. Specifically, previous literature demonstrates that learnable PEs typically underperform compared to deterministic ones. Additionally, in LLMs, the input sequence length can vary significantly and may exceed the maximum sequence length encountered during training, further complicating the use of learned embeddings.
> > >
> > > We hope these clarifications and planned revisions address your remaining concerns. Please do not hesitate to let us know if you have any further questions or concerns. We would be more than happy to provide additional clarification or engage in further discussion.
> > >
> > > Once again, we sincerely thank you for your valuable feedback, which will undoubtedly strengthen our work. We look forward to your further feedback, and we kindly ask that you consider these clarifications in your final grading.
> > >
> > > \
> > > Best regards,
> > >
> > > Authors
> > >
> > > \
> > > [1] Su, Jianlin, et al. "Roformer: Enhanced transformer with rotary position embedding." Neurocomputing 568 (2024): 127063.
> > >
> > > [2] Bai, Shuai, et al. "Qwen2. 5-vl technical report." arXiv preprint arXiv:2502.13923 (2025).

---

### Official Review · Reviewer_hPaZ · 2025-07-01

**Clarity:** 3
**Significance:** 3
**Originality:** 3
**Rating:** 4
**Confidence:** 4

**Summary:**

To address the challenge of effectively integrating images and structured data in multimodal time series forecasting, this paper proposes a Physics-Informed Position Encoding (PIPE) method. PIPE aims to better extract the underlying time and position information embedded in satellite images and enhance the alignment between visual data and time series. The method is applied to complex meteorological scenarios such as typhoon intensity and trajectory forecasting.

**Questions:**

1. Can PIPE be extended to other domains such as remote sensing in agriculture, urban heat island analysis, or sea ice prediction? Have similar tasks been attempted?

2. Have you conducted experiments to evaluate the performance of PIPE across different typhoon intensity levels? I suppose the performance may vary depending on the severity of the typhoon.

3. Apart from using negative indices, have you considered other mechanisms to separate visual and textual tokens (e.g., using token type embeddings)?

**Ethical Concerns:**

["NO or VERY MINOR ethics concerns only"]

**Final Justification:**

The authors promptly and thoroughly reply to my concerns, mainly solving the issues about the title specific for typhoon, more validation datasets and special validation on key components. Therefore, I prefer to increase my original rating to a positive one. However, I could not agree that "being up front about the limitations should be rewarded rather than punished“. It depends on whether the limitation is a key problem. I think for physical information, position encoding only occupies a very small part. That's why I could not increase more.

**Limitations:**

It would be helpful to further elaborate on the applicability of the proposed model—whether it is limited to weather-related tasks or potentially effective across a broader range of multimodal applications.

**Quality:**

2

**Strengths And Weaknesses:**

1. Advantages:

(1) This paper proposes the PIPE method as a targeted solution for effectively integrating sequential data and satellite imagery. While it is applied to typhoon prediction in this work, the approach has significant practical relevance.

(2) The proposed physics-informed positional indexing method distinguishes image patches from sequential data tokens using negative indices—a novel and elegant design that effectively addresses index conflicts.

(3) PIPE achieves state-of-the-art performance in both deep learning-based forecasting and climate modeling tasks.

2. Weaknesses :

(1) The main contributions of the paper focus on incorporating physical information through positional indexing and encoding at the model input level. Although the paper claims to be physics-informed, the physical guidance is limited to input encoding and does not extend to more meaningful components such as physical knowledge or physics-guided loss functions or constraints on model behavior.

(2) The title of this paper is for alignment of satellite images and time series, but the experiments are restricted to the domain of typhoon forecasting and only one testing dataset, which cannot fully support the main claim and limits the generalization of the proposed method.

(3) The paper does not investigate how PIPE performs across different categories of typhoon intensity. Since typhoons are typically classified into six levels, conducting experiments to analyze the model’s performance across these different intensity levels would provide deeper insights into its robustness and practical utility.

(4) If the evaluation remains limited to a single typhoon dataset, I suggest explicitly mentioning "typhoon" in the title to reflect the task more accurately. Moreover, if the focus is indeed on typhoon prediction, the method should be validated on additional typhoon datasets and compared with more typhoon-specific baselines.

---

> ### Author Rebuttal · Authors · 2025-07-31
>
> Dear Reviewer [hPaZ],
>
> Thank you for your efforts and comments.
>
> > Weakness #1: The main contributions of the paper focus on ... or constraints on model behavior.
>
> Thank you for your advice.
>
> 1. We would like to respectfully emphasize that, to the best of our knowledge, our method is the first to incorporate global physical information in a way that enables VLMs to effectively capture the visual features of satellite imagery, marking a significant step forward in this emerging research direction.
>
>     Our core contribution lies in the proposed **physics-informed position encoding**, which leverages spatiotemporal metadata (e.g., timestamps, latitude, and longitude) to inject physical context into the model. Broadening the scope to include additional physical constraints or features beyond positional encoding would have introduced substantial complexity and risked diluting the focus of this paper. Instead, we deliberately centered our efforts on thoroughly evaluating the effectiveness of our position encoding approach through comprehensive experiments and analysis. We recognize the importance of integrating broader physical laws and plan to pursue this promising direction in future work.
>
> 2. Furthermore, we have explicitly acknowledged this limitation and highlighted the integration of additional components (e.g., physical knowledge, physics-guided loss functions, or constraints on model behavior) as an important direction for future work (line 643 in the original manuscript). In line with the review guidelines, We believe that being up front about the limitations should be rewarded rather than punished.
>
> > Weakness #2: The title of this paper is for alignment of satellite images and time series, ... limits the generalization of the proposed method.
> >
> > Weakness #4: If the evaluation remains limited to a single typhoon dataset, I suggest explicitly mentioning "typhoon" in the title ... more typhoon-specific baselines.
> >
> > Question #1: Can PIPE be extended to other domains ... Have similar tasks been attempted?
> >
> > Limitation #1: It would be helpful to further elaborate on the applicability ... a broader range of multimodal applications.
>
> Thank you for your suggestion. To demonstrate the generalizability of our method, we added two additional experiments targeting different application domains: one focuses on typhoon forecasting in another region (the Australian (AU) region) [1]; the other uses the T31TFM-16 dataset for crop monitoring [2]. The first experiment confirms the robustness of our method in typhoon forecasting, while the second demonstrates its applicability to a different domain, showcasing its broader generalization capability.
>
> Though we validate the generalizability of our method by adding two additional experiments, we also would like to **emphasize typhoon forecasting in the title**, as you suggested. The new title will be ‘PIPE: Physics-Informed Position Encoding for Alignment of Satellite Images and Time Series of Typhoon Forecasting’
>
> 1. AU typhoon experiment:
>   * Settings: The settings remain the same as PIPE. The focus is also on typhoon forecasting, a regression task, which is the core of our paper.
>   * Result:
> |Models|Intensity MAE|Intensity RMSE|Latitude MAE|Latitude RMSE|Longitude MAE|Longitude RMSE|Distance MAE|
> |---|---|---|---|---|---|---|---|
> |Qwen-2.5-VL-3B (after tuning)|1.399|2.806|0.098|0.201|0.121|0.211|19.718|
> |PIPE-3B (Zero-shot)|1.361|2.773|0.094|0.171|0.118|0.206|19.090|
> |PIPE-3B (after tuning)|1.352|2.558|0.093|0.163|0.116|0.202|18.835|
>
>   * Analysis:
> The results show that our method can be generalized to different regions for typhoon forecasting. Even the model trained on the West Pacific region can perform well in the new dataset (zero-shot). For instance, compared to Qwen-2.5-VL-3B (after tuning), PIPE-3B (Zero-shot) achieved a lower intensity RMSE (2.773 vs. 2.806) and a smaller distance error (19.090 vs. 19.718). Additionally, PIPE-3B (after tuning) outperformed both models, achieving the best overall performance.
>
> 2. Crop monitoring experiment:
>   * Setting: We adopt our method to the semantic segmentation task. The training data and test data were split by the original dataset, with a ratio of 0.85:0.15. We calculate latitude and longitude from the given UTM information provided in the dataset. We use the ViT model to encode the satellite image patches and incorporate each patch’s physical information. The focus is on crop monitoring, a segmentation task. Number of epochs: 50. lr: 1e-4. Optimizer: Adam.
>
>   * Result:
> |Models|IoU|Accuracy|
> |---|---|---|
> |ViT PE|67.52|80.61|
> |3D PE|67.27|80.43|
> |PIPE|69.21|81.81|
>
>   * Analysis: The results show that our method achieves the best result (69.21% and 81.81%) than standard ViT PE (67.52% and 80.61%)  and 3D PE (67.27% and 80.43%). The experiment shows that our method can be generalized to other domains by incorporating global physical information.
>
> > Weakness #3:The paper does not investigate how PIPE performs across different categories of typhoon intensity. ... practical utility.
> >
> > Question #2: Have you conducted experiments ... depending on the severity of the typhoon.
>
> Thank you for your suggestion. We added the experiments on different intensity levels.
>
>   * Experiment settings:
> The settings remain the same as the experiment in the paper.
>
> After splitting, the dataset begins from Grade 2, and the distribution of samples is as follows:
> Grade 2: 18027.
> Grade 3: 17193.
> Grade 4: 8328.
> Grade 5: 10649.
> Grade 6: 15558.
>
>   * Result:
> |Grades|Intensity MAE|Intensity RMSE|Latitude MAE|Latitude RMSE|Longitude MAE|Longitude RMSE|Distance MAE|
> |---|---|---|---|---|---|---|---|
> |Grade 2|0.722|1.227|0.108|0.206|0.138|0.245|22.029|
> |Grade 3|0.997|1.772|0.097|0.169|0.115|0.200|18.902|
> |Grade 4|1.656|2.732|0.099|0.156|0.122|0.188|19.310|
> |Grade 5|2.707|4.556|0.090|0.163|0.116|0.203|18.418|
> |Grade 6|1.074|1.830|0.208|0.385|0.283|0.488|37.193|
> |All|1.515|2.981|0.084|0.159|0.095|0.178|16.275|
>
>   * Analysis:
>     * Definition of grades (helps to understand the results):
>
>       * Grades 3, 4, and 5 represent tropical cyclones, with Grade 5 being the most intense (not Grade 6).
>
>       * Grade 2 represents tropical depressions, which are weaker cyclones and not classified as tropical cyclones.
>
>       * Grade 6 represents a cyclone with a different structural system from tropical cyclones (not by tense). Typhoons typically transit to Grade 6 as their structure changes and then finish, instead of reverting to lower grades like 5, 4, 3, or 2.
>   * Results analysis:
>     * 1. Track Forecasting vs. Pressure Forecasting:
>       * Tropical cyclones (Grades 3, 4, 5) demonstrate better track forecasting performance compared to non-tropical cyclones (Grades 2, 6). This is likely due to the more stable and predictable system of tropical cyclones, which facilitates track forecasting. However, pressure forecasting is more challenging for tropical cyclones due to higher variability and complexity.
>       * In contrast, non-tropical cyclones (Grade 2 and Grade 6) exhibit more predictable pressure patterns but greater difficulty in track forecasting. This is especially pronounced before the cyclone structure forms (Grade 2) and after it breaks (Grade 6).
>     * 2. Impact of Intensity on Forecasting:
>       * As the intensity of tropical cyclones increases (Grades 3, 4, 5), pressure forecasting becomes progressively more challenging. However, the difficulty of track forecasting remains relatively consistent across these grades.
>     * 3. Impact of Data Volume:
>       * Models trained on all grades perform better in track forecasting compared to models trained on individual grades. This result indicates that a larger volume of training data improves track forecasting accuracy across different cyclone intensities.
>
> > Question #3: Apart from using negative indices, have you considered other mechanisms to separate visual and textual tokens (e.g., using token type embeddings)?
>
> Thanks for the suggestion.
> 1. We had explored an alternative mechanism, **offset indices**, where all image tokens are assigned an offset equal to the maximum length of the textual tokens. The results of this experiment are summarized in the table below:
> |Model|Intensity MAE|Intensity RMSE|Latitude MAE|Latitude RMSE|Longitude MAE|Longitude RMSE|Distance MAE|
> |---|---|---|---|---|---|---|---|
> |offset|1.815|3.762|0.188|0.303|0.229|0.385|30.157|
> |negative|1.515|2.981|0.084|0.159|0.095|0.178|16.275|
>
>     Based on this experiment, we found that our choice of using negative indices is an intuitive and efficient method for separating different types of tokens.
>
> 2. Regarding using token type embeddings, we did not find evidence of their application in existing VLMs (e.g., Qwen [3], LLaVA [4], ViT [5], etc.). Moreover, the use of negative indices for image tokens inherently provides a natural and effective way to distinguish between visual and textual token types without additional complexity.
>
> \
> Finally, we want to thank you once again for your thoughtful comments. We look forward to your feedback and we kindly ask that you consider these clarifications in your final grading.
>
> Warm regards,
>
> The Authors
>
> \
> [1] Kitamoto, Asanobu, Erwan Dzik, and Gaspar Faure. "Machine Learning for the Digital Typhoon Dataset: Extensions to Multiple Basins and New Developments in Representations and Tasks." arXiv preprint arXiv:2411.16421 (2024).
>
> [2] Tarasiou, Michail, Riza Alp Güler, and Stefanos Zafeiriou. "Context-self contrastive pretraining for crop type semantic segmentation." IEEE Transactions on Geoscience and Remote Sensing 60 (2022): 1-17.
>
> [3] Bai, Shuai, et al. "Qwen2. 5-vl technical report." arXiv preprint arXiv:2502.13923 (2025).
>
> [4] Liu, Haotian, et al. "Visual instruction tuning." Advances in neural information processing systems 36 (2023): 34892-34916.
>
> [5] Dosovitskiy, Alexey, et al. "An image is worth 16x16 words: Transformers for image recognition at scale." arXiv preprint arXiv:2010.11929 (2020).

---

> > ### Author Response · Authors · 2025-08-05
> >
> > Dear Reviewer [hPaZ],
> >
> > Thank you once again for your thoughtful review. To address your concerns regarding the contribution, generalizability, and performance across different intensity levels, we have conducted four additional experiments and provided further clarifications.
> >
> > As the discussion deadline approaches, please feel free to reach out with any further questions or if additional clarification is needed. We would be more than happy to engage in further discussion. If you find that we have addressed your concerns, we kindly request your consideration in improving our score. Thank you very much.
> >
> > \
> > Best regards,
> >
> > Authors

---

> > > ### Comment · Reviewer_hPaZ · 2025-08-05
> > >
> > > Thanks for addressing my concerns. Sorry for replying late.
> > > The supplemented experiments make this work more convincing.
> > > For the new added two datasets [1-2], what are the results of traditional methods?
> > > No more experiments are needed. I think the resuts are already contained in previous works. Please give them like Table 1 in the main manuscript. I would like to have a sense of the performance comparison with previous works.
> > > Thanks.

---

> > > > ### Author Response · Authors · 2025-08-05
> > > >
> > > > Dear Reviewer [hPaZ],
> > > >
> > > > Thank you for your response. We are pleased that our four additional experiments and justifications addressed your concerns. Below, we provide the results of traditional methods for the two newly added datasets, as requested.
> > > >
> > > > 1. AU Typhoon (regression task, the core of our paper)
> > > >
> > > > |Models|Intensity MAE|Intensity RMSE|Latitude MAE|Latitude RMSE|Longitude MAE|Longitude RMSE|Distance MAE|
> > > > |---|---|---|---|---|---|---|---|
> > > > |TCN|1.613|---|---|---|---|---|30.588|
> > > > |ECMWF-HRES|---|---|---|---|---|---|27.181|
> > > > |PanGu|---|---|---|---|---|---|32.892|
> > > > |GenCast|---|---|---|---|---|---|20.331|
> > > > |TIFS|---|7.292|---|---|---|---|---|
> > > > |PatchTST|1.784|2.651|0.169|0.231|0.272|0.367|38.056|
> > > > |iTransformer|1.556|2.588|0.164|0.225|0.213|0.302|32.412|
> > > > |Crossformer|3.047|4.443|0.400|0.536|0.920|1.254|116.994|
> > > > |TiDE|1.464|2.538|0.153|0.210|0.206|0.292|30.808|
> > > > |One Fits All|1.499|2.534|0.170|0.234|0.236|0.323|34.932|
> > > > |AutoTimes|1.617|2.656|0.188|0.256|0.266|0.363|39.038|
> > > > |TimesNet|2.987|4.607|0.245|0.334|0.979|1.299|112.211|
> > > > |TimeMixer|1.565|2.548|0.161|0.218|0.237|0.312|34.157|
> > > > |Qwen-2.5-VL-3B (after tuning)|1.399|2.806|0.098|0.201|0.121|0.211|19.718|
> > > > |PIPE-3B (Zero-shot)|1.361|2.773|0.094|0.171|0.118|0.206|19.090|
> > > > |PIPE-3B (after tuning)|1.352|2.558|0.093|0.163|0.116|0.202|18.835|
> > > >
> > > > Analysis: Our method achieves SOTA performance among traditional methods. Even in the zero-shot setting, it performs competitively (e.g., for distance error: 19.090km, performing better than all other methods). This highlights the effectiveness of incorporating global physical information, enabling the model to generalize to other regions automatically for typhoon forecasting.
> > > >
> > > > 2. Crop monitoring (semantic segmentation task)
> > > > |Models|IoU|Accuracy|
> > > > |---|---|---|
> > > > |ViT model with ViT PE|67.52|80.61|
> > > > |ViT model with 3D PE|67.27|80.43|
> > > > |ViT model with PIPE|69.21|81.81|
> > > > |SOTA by the original paper|74.20|85.19|
> > > >
> > > > Analysis:
> > > > Semantic segmentation is another research direction, and the original paper employs sophisticated models specifically designed for the semantic segmentation task. In contrast, we use a simple ViT model to demonstrate the effectiveness of our PIPE method, constrained by time and resource limitations. The results show that **PIPE improves ViT's performance** (69.21 IoU vs. 67.27 IoU, 81.81 Accuracy vs. 80.43 Accuracy). While the best ViT with PIPE does not surpass the SOTA models specifically designed for this semantic segmentation task, it can be inferred that applying PIPE to the same sophisticated model used in the original paper could further enhance SOTA performance since we have proved that **PIPE enhances both ViT (simple vision model) and Qwen models (Multimodal LLM)**.
> > > >
> > > > \
> > > > We hope these results and analyses address your request. Should you have any further questions, please feel free to let us know. We would be glad to provide additional clarification or engage in further discussion.
> > > >
> > > > Once again, we sincerely thank you for your valuable feedback, which will undoubtedly strengthen our work. We kindly request your consideration in improving our score if your questions are addressed.
> > > >
> > > > \
> > > > Best regards,
> > > >
> > > > Authors

---

> > > > > ### Comment · Reviewer_hPaZ · 2025-08-06
> > > > >
> > > > > Thanks for the prompt reply.

---

> ### Author Response · Authors · 2025-08-06
>
> Dear Reviewer [hPaZ],
>
> Thank you for your response. We are pleased that **we have addressed all your concerns**. We will incorporate these revisions into the final version of the paper. We would be **truly grateful if you could consider increasing your recommendation to a higher level**, as your suggested revisions have significantly enhanced the quality of the paper.
>
> Thank you once again for your valuable advice.
>
> \
> Best regards,
>
> Authors

---

### Official Review · Reviewer_8J4Q · 2025-07-03

**Clarity:** 3
**Significance:** 2
**Originality:** 3
**Rating:** 4
**Confidence:** 4

**Summary:**

This paper proposes PIPE (Physics-Informed Positional Encoding), a multimodal time-series forecasting framework that integrates visual information using a VLM. The key contribution is its physics-informed positional encoding, which embeds real-world metadata (latitude, longitude, and timestamp) into visual token positions, enabling better spatiotemporal alignment across inputs. A variant-frequency sinusoidal function further captures the periodic nature of physical variables. PIPE is evaluated on typhoon forecasting, showing that incorporating physical context improves multimodal prediction.

**Questions:**

See the comments above.

Additionally, why does the Figure 1 show two distinct points with the same timestamp (2018/10/29 00:00) but different latitude, longitude, and intensity values?

**Ethical Concerns:**

["NO or VERY MINOR ethics concerns only"]

**Final Justification:**

The authors have provided more results, which have partially addressed my concerns.

**Limitations:**

Have the authors adequately addressed the limitations and potential negative societal impact of their work? If so, simply leave “yes”; if not, please include constructive suggestions for improvement. In general, authors should be rewarded rather than punished for being up front about the limitations of their work and any potential negative societal impact. You are encouraged to think through whether any critical points are missing and provide these as feedback for the authors.

**Quality:**

2

**Strengths And Weaknesses:**

Strengths
1.	The paper proposes a lightweight method to embed physical information into VLMs, allowing the model to encode shared global physical knowledges in a simple way. This approach has potential benefits for a wide range of transformer-based models in time-series prediction tasks.
2.	The framework makes effective use of historical physical metadata (e.g., latitude, longitude, and time): once as explicit values in the textual input, and again through physics-informed positional indexing applied to visual tokens.
3.	The application to typhoon forecasting is highly relevant and impactful, with potential real-world benefits for disaster preparedness and evacuation planning.
4.	The paper is clearly structured, and easy to follow.

Weakness
1.	The first claimed contribution may be overstated. While the paper introduces a novel positional encoding scheme, the overall architecture closely follows the standard structure of VLMs for multimodal input processing.
2.	The comparison against baselines is potentially unfair. Most baseline models do not incorporate visual data, except for Qwen-2.5-VL-3B. Since the performance gains over Qwen-2.5-VL-3B are relatively small, the strength of the contribution may be overstated without broader comparison to other vision-enabled baselines.
3.	While the proposed method is demonstrated on typhoon forecasting, other real-world applications could benefit from embedding physical information. If the results in the typhoon domain cannot be reliably reproduced, it would be valuable for the authors to evaluate the method in additional domains to support its generalizability.
4.	Minor typographical issue: in line 141, there should be a space between “encoding” and “PIPE (PIPE).”

---

> ### Author Rebuttal · Authors · 2025-07-31
>
> Dear Reviewer [8J4Q],
>
> Thank you for your time and feedback.
>
> > Weakness #1: The first claimed contribution may be overstated. While the paper introduces a novel positional encoding scheme, the overall architecture closely follows the standard structure of VLMs for multimodal input processing.
>
> Thank you for your advice on the writing. We will revise the statement as suggested to ‘We propose a novel positional encoding **scheme** that …’ to ensure the wording is accurate and not overstated.
>
> Meanwhile, we would like to respectfully clarify that the essence of our first contribution remains strong and valid. To our best knowledge, we are the first to integrate global physical information into position encoding to capture visual features.  While existing approaches primarily focus on incorporating textual or tabular data in time series forecasting, our method opens a new direction by enabling VLMs to effectively leverage physical context through position encoding.
>
> > Weakness #2: The comparison against baselines is potentially unfair. Most baseline models do not incorporate visual data, except for Qwen-2.5-VL-3B. Since the performance gains over Qwen-2.5-VL-3B are relatively small, the strength of the contribution may be overstated without broader comparison to other vision-enabled baselines.
>
> Thank you for your comment. We have tried our best to cover the majority of available baselines, including domain-specific models and machine learning models (both vision-based and non-vision-based). This comprehensive evaluation is also appreciated by Reviewer [NJNg], who noted that we “have performed rigorous experiments.” However, due to the limitations of existing methods, many of them did not incorporate vision data for time series forecasting, which further highlights the novelty of our approach.
>
> Furthermore, we would like to respectfully argue that:
> 1. Besides Qwen-2.5-VL-3B, we also have included other benchmark results in the original manuscript from the dataset benchmarking, which incorporates visual data. For instance, in **Table 4**, the 12-hour result is included for comparison (only the 12-hour result is available in the original benchmarking).
> 2. To further enhance the vision-enabled baselines and demonstrate the effectiveness of our method, besides the open-source Qwen-2.5-VL-3B, we added one additional experiment on closed-source MLLM: gemini-2.5-flash, which also incorporates visual data.
>   * Experiment setting: the input remains the same settings as Qwen and PIPE.
>   * Result:
> |Models|Intensity MAE|Intensity RMSE|Latitude MAE|Latitude RMSE|Longitude MAE|Longitude RMSE|Distance MAE|
> |---|---|---|---|---|---|---|---|
> |gemini-2.5-flash|1.924|3.654|0.174|0.254|0.237|0.798|36.006|
> |PIPE|1.515|2.981|0.084|0.159|0.095|0.178|16.275|
>   * Analysis: The results show that our method can be **more effective** in integrating satellite visual features than existing multimodal methods.
>
> 3. We believe that integrating visual data represents our unique contribution, addressing a significant research gap in multimodal forecasting methods. The existing approaches are limited since they mainly align numeric (and text) data. We respectfully believe this should be seen as a strength rather than a weakness.
> 4. Regarding the gain over Qwen-2.5-VL-3B, our method achieved: 6.7% (intensity MAE)，8.4% (intensity RMSE),  3.5% (lat MAE), 1.9% (lat RMSE), 8.4%(lng MAE), 5.1%(lng RMSE), and 5.2% (distance). We believe that these results are able to demonstrate the effectiveness of our proposed method.
>
> > Weakness #3: While the proposed method is demonstrated on typhoon forecasting, other real-world applications could benefit from embedding physical information. If the results in the typhoon domain cannot be reliably reproduced, it would be valuable for the authors to evaluate the method in additional domains to support its generalizability.
>
> Thank you for your suggestion. To demonstrate the generalizability of our method, we added two additional experiments targeting different application domains: one focuses on typhoon forecasting in another region (the Australian (AU) region) [1]; the other uses the T31TFM-16 dataset for crop monitoring [2]. The first experiment confirms the robustness of our method in typhoon forecasting, while the second demonstrates its applicability to a different domain, showcasing its broader generalization capability.
>
> 1. AU typhoon experiment:
>   * Settings: The settings remain the same as PIPE. The focus is on typhoon forecasting, a regression task, which is the core of our paper.
>   * Result:
> |Models|Intensity MAE|Intensity RMSE|Latitude MAE|Latitude RMSE|Longitude MAE|Longitude RMSE|Distance MAE|
> |---|---|---|---|---|---|---|---|
> |Qwen-2.5-VL-3B (after tuning)|1.399|2.806|0.098|0.201|0.121|0.211|19.718|
> |PIPE-3B (Zero-shot)|1.361|2.773|0.094|0.171|0.118|0.206|19.090|
> |PIPE-3B (after tuning)|1.352|2.558|0.093|0.163|0.116|0.202|18.835|
>
>   * Analysis:
> The results show that our method can be generalized to different regions for typhoon forecasting. Even the model trained on the West Pacific region can perform well in the new dataset (zero-shot). For instance, compared to Qwen-2.5-VL-3B (after tuning), PIPE-3B (Zero-shot) achieved a lower intensity RMSE (2.773 vs. 2.806) and a smaller distance error (19.090 vs. 19.718). Additionally, PIPE-3B (after tuning) outperformed both models, achieving the best overall performance.
>
> 2. Crop monitoring experiment:
>   * Setting: We adopt our method to the semantic segmentation task. The training data and test data were split by the original dataset, with a ratio of 0.85:0.15. We calculate latitude and longitude from the given UTM information provided in the dataset. We use the ViT model to encode the satellite image patches and incorporate each patch’s physical information. The focus is on crop monitoring, a segmentation task. Number of epochs: 50. lr: 1e-4. Optimizer: Adam.
>
>   * Result:
> |Models|IoU|Accuracy|
> |---|---|---|
> |ViT PE|67.52|80.61|
> |3D PE|67.27|80.43|
> |PIPE|69.21|81.81|
>
>   * Analysis: The results show that our method achieves the best result (69.21% and 81.81%) than standard ViT PE (67.52% and 80.61%)  and 3D PE (67.27% and 80.43%). The experiment shows that our method can be generalized to other domains by incorporating global physical information.
>
> > Weakness #4: Minor typographical issue: in line 141, there should be a space between “encoding” and “PIPE (PIPE).”
> >
> > Question #1: why does the Figure 1 show two distinct points with the same timestamp (2018/10/29 00:00) but different latitude, longitude, and intensity values?
>
> Thank you for your feedback. We will correct the typos.
>
> \
> Finally, we want to thank you once again for your thoughtful comments. We look forward to your feedback and we kindly ask that you consider these clarifications in your final grading.
>
> Warm regards,
>
> The Authors
>
> \
> [1] Kitamoto, Asanobu, Erwan Dzik, and Gaspar Faure. "Machine Learning for the Digital Typhoon Dataset: Extensions to Multiple Basins and New Developments in Representations and Tasks." arXiv preprint arXiv:2411.16421 (2024).
>
> [2] Tarasiou, Michail, Riza Alp Güler, and Stefanos Zafeiriou. "Context-self contrastive pretraining for crop type semantic segmentation." IEEE Transactions on Geoscience and Remote Sensing 60 (2022): 1-17.

---

> ### Author Response · Authors · 2025-08-05
>
> Dear Reviewer [8J4Q],
>
> Thank you once again for your thoughtful review. To address your concerns regarding the claimed contributions, baselines, and generalizability, we have added three additional experiments and provided further clarifications.
>
> As the discussion deadline approaches, please feel free to reach out with any further questions or if additional clarification is needed. We would be more than happy to engage in further discussion.
> If you find that we have addressed your concerns, we would be **truly grateful if you could consider increasing your recommendation to a higher level**.
>
> \
> Best regards,
>
> Authors

---

### Official Review · Reviewer_NJNg · 2025-07-04

**Clarity:** 2
**Significance:** 2
**Originality:** 2
**Rating:** 5
**Confidence:** 4

**Summary:**

This paper introduces PIPE (Physics-Informed Position Encoding), a framework for multimodal time series forecasting, with a focus on integrating satellite imagery and numerical data for tasks such as typhoon prediction. Unlike existing models that either ignore visual data or only capture pixel-level semantics, PIPE incorporates physical metadata (timestamps, latitude, longitude) directly into the position encoding of vision-language models (VLMs).
PIPE is evaluated on Typhoon dataset and achieves state-of-the-art forecasting accuracy, notably improving typhoon intensity and track predictions compared to both AI and domain-specific baselines.

**Questions:**

Please see weakness, if you can address them

**Ethical Concerns:**

["NO or VERY MINOR ethics concerns only"]

**Final Justification:**

The authors have given satisfactory reply to all my questions and given additional results as well. I have increased their ratings.

**Limitations:**

One paragraph on limitations of the work should be included

**Paper Formatting Concerns:**

No concern

**Quality:**

3

**Strengths And Weaknesses:**

Strengths
Introducing PIPE which explicitly embeds physical attributes—like timestamps, latitude, and longitude—into both the positional indexing and encoding schemes of a vision-language forecasting model. This is beyond traditional positional encoding (which only preserves token order) by incorporating global, real-world geospatial and temporal information directly into the model’s structure. This aligns satellite imagery with time-series data, enabling the model to understand not just visual patterns but also their physical context, crucial for accurate climate forecasting. Authors have performed rigorous experiments to establish their proposals.

Weaknesses
It has been tested only on one domain, experiments could be performed on other applications such as crop monitoring, climate change, etc.

Although, the title of the paper is physics informed…, it does not modal physics laws, constraints etc.

While PIPE innovatively encodes timestamps and geo-coordinates (latitude & longitude), it does not incorporate other rich physical attributes (like atmospheric pressure fields, wind vectors, or physical constraints from fluid dynamics).

The authors themselves note future work should integrate broader physical laws and constraints to better model real-world dynamics.

While PIPE improves short (6–12 hour) forecasts, I am not sure how does it analyze very long-term forecasting performance (e.g., multiple days, weeks ahead) where errors compound and encoding choices might have more dramatic effects.

Authors could have included the comparison between standard positional encodings and variant -frequency positional encodings.

Few observations:

The statement in the abstract: “However, existing multimodal approaches primarily ….., leaving the visual data in existing time series datasets untouched.” The integration of time series and vision (multimodal) data is discussed in the following works: Paper 1, Paper 2. But the wording is overly absolute.

In Figure 1, the axes and legend require clarification. The units for intensity are not explicitly labelled (e.g., hPa)
In Figures 4 and 5, the x-axis, y-axis, and legend are not properly visible.
Algorithm 1 and related descriptions are insightful, but formatting inconsistencies (e.g., indentation of bullet points, alignment of variables) make it hard to follow. Please refine for clarity.

---

> ### Author Rebuttal · Authors · 2025-07-31
>
> Dear Reviewer [NJNg],
>
> Thank you for your efforts and positive feedback.
>
> > Weakness #1: It has been tested only on one domain, experiments could be performed on other applications such as crop monitoring, climate change, etc.
>
> Thank you for your suggestion. To demonstrate the generalizability of our method, we added two additional experiments targeting different application domains: one focuses on typhoon forecasting in another region (the Australian (AU) region) [1]; the other uses the T31TFM-16 dataset for crop monitoring [2]. The first experiment confirms the robustness of our method in typhoon forecasting, while the second demonstrates its applicability to a different domain, showcasing its broader generalization capability.
>
> **1. AU typhoon experiment:**
>   * Setting: The settings remain the same as PIPE. The focus is on typhoon forecasting, a regression task, which is the core of our paper.
>   * Result:
> |Models|Intensity MAE|Intensity RMSE|Latitude MAE|Latitude RMSE|Longitude MAE|Longitude RMSE|Distance MAE|
> |---|---|---|---|---|---|---|---|
> |Qwen-2.5-VL-3B (after tuning)|1.399|2.806|0.098|0.201|0.121|0.211|19.718|
> |PIPE-3B (Zero-shot)|1.361|2.773|0.094|0.171|0.118|0.206|19.090|
> |PIPE-3B (after tuning)|1.352|2.558|0.093|0.163|0.116|0.202|18.835|
>   * Analysis: The results show that our method can be generalized to different regions for typhoon forecasting. Even the model trained on the West Pacific region can perform well in the new dataset (zero-shot). For instance, compared to Qwen-2.5-VL-3B (after tuning), PIPE-3B (Zero-shot) achieved a lower intensity RMSE (2.773 vs. 2.806) and a smaller distance error (19.090 vs. 19.718). Additionally, PIPE-3B (after tuning) outperformed both models, achieving the best overall performance.
>
> **2. Crop monitoring experiment:**
>
>   * Setting: We adopt our method to the semantic segmentation task. The training data and test data were split by the original dataset, with a ratio of 0.85:0.15. We calculate latitude and longitude from the given UTM information provided in the dataset. We use the ViT model to encode the satellite image patches and incorporate each patch’s physical information. The focus is on crop monitoring, a segmentation task. Number of epochs: 50. lr: 1e-4. Optimizer: Adam.
>
>   * Result:
> |Models|IoU|Accuracy|
> |---|---|---|
> |ViT PE|67.52|80.61|
> |3D PE|67.27|80.43|
> |PIPE|69.21|81.81|
>
>   * Analysis: The results show that our method achieves the best result (69.21% and 81.81%) than standard ViT PE (67.52% and 80.61%)  and 3D PE (67.27% and 80.43%). The experiment shows that our method can be generalized to other domains by incorporating global physical information.
>
> > Weakness #2: Although, the title of the paper is physics informed…, it does not modal physics laws, constraints etc. While PIPE innovatively encodes timestamps and geo-coordinates (latitude & longitude), it does not incorporate other rich physical attributes (like atmospheric pressure fields, wind vectors, or physical constraints from fluid dynamics). The authors themselves note future work should integrate broader physical laws and constraints to better model real-world dynamics.
>
> Thank you for your advice.
>
> 1. We would like to respectfully emphasize that, to the best of our knowledge, our method is the first to incorporate global physical information in a way that enables VLMs to effectively capture the visual features of satellite imagery, marking a significant step forward in this emerging research direction.
> Our core contribution lies in the proposed **physics-informed position encoding**, which leverages spatiotemporal metadata (e.g., timestamps, latitude, and longitude) to inject physical context into the model. Broadening the scope to include additional physical constraints or features beyond positional encoding would have introduced substantial complexity and risked diluting the focus of this paper. Instead, we deliberately centered our efforts on thoroughly evaluating the effectiveness of our position encoding approach through comprehensive experiments and analysis. We recognize the importance of integrating broader physical laws and plan to pursue this promising direction in future work.
>
> 2. Furthermore, we have explicitly acknowledged this limitation and highlighted the integration of additional physical attributes and constraints (e.g., atmospheric pressure, wind vectors, and fluid dynamics laws) as an important direction for future work (line 643 in the original manuscript). In line with the review guidelines, We believe that being up front about the limitations should be rewarded rather than punished.
>
> > Weakness #3: While PIPE improves short (6–12 hour) forecasts, I am not sure how does it analyze very long-term forecasting performance (e.g., multiple days, weeks ahead) where errors compound and encoding choices might have more dramatic effects.
>
> Thank you for your feedback.
>
> 1. **Short-term forecasting is of critical importance** for timely decision-making, as typhoon events typically span only a few days. For example, the maximum forecasting length is limited to 12 hours in the dataset baseline used in our study and limited to 1 hour in [3], which aligns with the practical need for short-term predictions.
> 2. We have also included the extended forecasting horizons for VLMs in the Limitations section (line 642). Addressing the challenge of long-term forecasting will be one of the key focuses of future work, particularly addressing the limitation of current VLMs to handle more input images. However, the **primary goal of this paper is to explore whether PIPE is effective** in incorporating additional physical information into forecasting models. We believe that our experimental results successfully demonstrate the validity of our methods.
>
> > Weakness #4: Authors could have included the comparison between standard positional encodings and variant -frequency positional encodings.
>
> Thank you for the valuable advice. We added the experiment as suggested:
>
> |Models|Intensity MAE|Intensity RMSE|Latitude MAE|Latitude RMSE|Longitude MAE|Longitude RMSE|Distance MAE|
> |---|---|---|---|---|---|---|---|
> |stantard PE|1.617|3.231|0.087|0.162|0.103|0.187|17.129|
> |only variant-frequency PE|1.581|2.994|0.087|0.161|0.102|0.186|17.117|
>
> The results show that the variant-frequency component can improve the performance over the standard positional encodings, since it encodes the physical attributes of the variables.
>
> > Observation #1: The statement in the abstract: “However, existing multimodal approaches primarily ….., leaving the visual data in existing time series datasets untouched.” The integration of time series and vision (multimodal) data is discussed in the following works: Paper 1, Paper 2. But the wording is overly absolute.
>
> Thank you for your advice. We agree that the current phrasing is not the most appropriate, and we will revise the statement to:  ‘leaving the visual data in existing time series datasets **underexplored**’.
> We would like to respectfully state that to our best knowledge, we are the first to integrate global physical information into position encoding to capture visual features. The existing methods focus on the incorporation of textual data in time series forecasting.
> We notice that you left two placeholders for existing works toward this research direction. We would greatly appreciate it if you could share these references with us to further improve our work.
>
> > Observation #2: In Figure 1, the axes and legend require clarification. The units for intensity are not explicitly labelled (e.g., hPa) In Figures 4 and 5, the x-axis, y-axis, and legend are not properly visible. Algorithm 1 and related descriptions are insightful, but formatting inconsistencies (e.g., indentation of bullet points, alignment of variables) make it hard to follow. Please refine for clarity.
>
> Thank you for the positive feedback on our algorithm and the advice to improve the presentation quality.
> In Figure 1, the x-axis is the longitude and the y-axis is the latitude. We will add hPa to the intensity.
> In Figures 4 and 5, we will increase the font size of the x-axis, y-axis, and legend.
> We will revise the format of Algorithm 1.
>
> > Limitations #1: One paragraph on limitations of the work should be included
>
> We have included the limitation in Appendix G (line 638). We will move them into the main section.
>
> \
> Finally, we want to thank you once again for your thoughtful comments. We look forward to your feedback and we kindly ask that you consider these clarifications in your final grading.
>
> Warm regards,
>
> The Authors
>
> \
> [1] Kitamoto, Asanobu, Erwan Dzik, and Gaspar Faure. "Machine Learning for the Digital Typhoon Dataset: Extensions to Multiple Basins and New Developments in Representations and Tasks." arXiv preprint arXiv:2411.16421 (2024).
>
> [2] Tarasiou, Michail, Riza Alp Güler, and Stefanos Zafeiriou. "Context-self contrastive pretraining for crop type semantic segmentation." IEEE Transactions on Geoscience and Remote Sensing 60 (2022): 1-17.
>
> [3] Zhu, Jiakai, and Jianhua Dai. "A rain-type adaptive optical flow method and its application in tropical cyclone rainfall nowcasting." Frontiers of Earth Science 16.2 (2022): 248-264.

---

> > ### Comment · Reviewer_NJNg · 2025-08-03
> > **PIPE: Physics-Informed Position Encoding for Alignment of Satellite Images and Time Series**
> >
> > I have read the response given by the authors and I will keep my positive ratings.

---

> > > ### Author Response · Authors · 2025-08-03
> > > **Reply to Reviewer NJNg's Comments on Rebuttal**
> > >
> > > Thank you very much for your thoughtful response and for acknowledging the detailed justifications we provided. We have taken great care to address all the concerns raised through extensive additional experiments and corresponding analyses, and we believe that the revised paper makes a meaningful contribution to the field. If you agree that our revisions have met the expectations for a stronger endorsement, we would be sincerely grateful for your consideration of an upgraded recommendation.
> > >
> > > Please do not hesitate to let us know if you have any further questions or concerns—we would be more than happy to provide additional clarification or engage in further discussion.

---

> > > ### Comment · Reviewer_NJNg · 2025-08-07
> > >
> > > I would like to thank authors for providing additional results. I have two observations- please clarify in case you feel otherwise.
> > > 1. About the response of Physics informed query--Qwen-2.5-v1 LLM like most general-purpose LLMs including GPT-4, LLaMA-3, etc. does not inherently have geospatial reasoning or map-based knowledge. So, how providing latitude and longitude automatically lead to more informative or contextual responses. are you providing real time look-up to maps etc.
> > > 2. There is hardly difference in numbers except intensity based metrics when you use variant frequency positional embeddings. What is the overhead of this encoding. is it worth using it?

---

> > > > ### Author Response · Authors · 2025-08-07
> > > >
> > > > Dear Reviewer [NJNg],
> > > >
> > > > Thank you very much for your feedback and further discussions.
> > > >
> > > > > About the response of Physics informed query--Qwen-2.5-v1 LLM like most general-purpose LLMs including GPT-4, LLaMA-3, etc. does not inherently have geospatial reasoning or map-based knowledge. So, how providing latitude and longitude automatically lead to more informative or contextual responses. are you providing real time look-up to maps etc.
> > > >
> > > > Thank you for your insights. We agree that general-purpose LLMs do not inherently have geospatial reasoning or map-based knowledge. However, **that is indeed why we incorporate additional physical information into the LLMs** during the post-training to contribute to this significant problem.
> > > >
> > > > 1. Limitation of existing methods.
> > > >
> > > >     PEs assign indices to tokens without distinguishing modalities. For example:
> > > >     * Input: [Text: "It is a dog"] + [Image Patches: P₁, P₂, ..., Pₙ]
> > > >     * PEs: [0, 1, 2, 3, 4, ..., n+3]
> > > >
> > > >     This treats text tokens (e.g., dog=position 3) and image patches (e.g., P₁=position 4) as homogeneous sequential units. Crucially, it fails to encode the physical relationships between satellite image patches (e.g., spatial proximity between P₁ and P₂) or their geospatial context.
> > > >
> > > > 2. Our Novel PE Scheme to address the limitation.
> > > >
> > > >     Our method incorporates the physical knowledge into the model. In scenarios involving satellite imagery, the physical information encoded in image patches is highly relevant to the output. By training with historical data, the model **captures global physical knowledge that remains consistent across input samples**. For example, in two different typhoons, the pattern of the typhoon eye might appear in the first image patch. Instead of assigning a generic PE (e.g., position 4) to both, our scheme incorporates more information, such as (t₁, lat₁, lng₁) and (t₂, lat₂, lng₂), and their physical attributes, such as frequency.
> > > >     This incorporation of geospatially grounded data allows the model to contextualize its responses based on real-world physics. As a result, providing latitude and longitude enables the model to generate outputs that are not only more contextual but also better informed by the physical relationships within the data.
> > > >
> > > >
> > > > > There is hardly difference in numbers except intensity based metrics when you use variant frequency positional embeddings. What is the overhead of this encoding. is it worth using it?
> > > >
> > > > Thank you for your feedback. We found that it is worth using it for two reasons:
> > > >
> > > > 1. The whole ablation study suggests that **the combination of both physics-informed indexing and variant frequency positional embeddings yields the most significant performance boost**.
> > > >
> > > >     In the ablation study you suggested, we compared variant frequency positional embeddings with the existing method. While the improvements are most significant for intensity-based metrics, there is also a noticeable (17.117 vs. 17.129) improvement in location-based metrics.
> > > >     Furthermore, when looking at the entire ablation study, the largest gains come from integrating both physics-informed indexing and variant frequency positional embeddings. For example, in terms of location error:
> > > >     * 16.275 (PIPE: both).
> > > >     * 16.860 (only physics-informed indexing).
> > > >     * 17.117 (only variant frequency positional embeddings).
> > > >     * 17.129 (existing method).
> > > >
> > > >     These results suggest that their combination yields the most significant performance boost. Therefore, we believe it is necessary to include variant frequency positional embeddings.
> > > >
> > > > 2. It is **computationally efficient**.
> > > >
> > > > The algorithm for variant frequency positional embeddings is non-parametric and, thus, very efficient. It is based on sinusoidal functions, which are computationally inexpensive and were a key reason for their adoption in the original Transformer architecture. Given their low computational cost and the additional performance gains they provide, we believe it is reasonable to use it.
> > > >
> > > > \
> > > > Thank you for your further insights and discussion. We will ensure that these insights and revisions are incorporated into the final version of the paper. We would be **truly grateful if you could consider increasing your recommendation to a higher level**, as your insights have been valuable in enhancing the quality of our work.
> > > >
> > > > Thank you once again for your valuable comments and suggestions.
> > > >
> > > > \
> > > > Best regards,
> > > >
> > > > Authors

---

> > ### Comment · Reviewer_NJNg · 2025-08-07
> >
> > Thank you for authors' response. I do not have any further questions.

---

> > > ### Author Response · Authors · 2025-08-07
> > >
> > > Dear Reviewer [NJNg],
> > >
> > > We are delighted that **all your questions have been successfully addressed**. We sincerely appreciate your questions and insights during the rebuttal stage.
> > >
> > > we would be **deeply grateful if you could consider increasing your recommendation to a higher level.**
> > >
> > > \
> > > Best regards,
> > >
> > > Authors

---

### Note · Authors · 2025-08-12

Dear AC,

We sincerely appreciate the reviewers’ thoughtful feedback. We also thank you for your efforts in reviewing our work. Below, we provide a summary for your consideration.

We are encouraged that the reviewers found our method to be:
* Novel and elegant design. Reviewer [NJNg, 8J4Q, hPaZ, iSKv]
* Impactful for real-world applications. Reviewer [NJNg, 8J4Q, hPaZ, iSKv]
* Lightweight and beneficial for a wide range of uses. Reviewer [8J4Q, iSKv]
* Effective for domain-specific tasks. Reviewer [NJNg, 8J4Q, hPaZ, fqdc]

They commended the paper to be:
* Rigorous experiments. Reviewer [NJNg, fqdc]
* Well-motivated. Reviewer [fqdc]
* Clearly structured. Reviewer [8J4Q ]

---
We have incorporated many helpful suggestions:
1. Enhanced Validation:
* Added two additional datasets and experiments to prove the generalizability. Reviewer. [NJNg, 8J4Q, hPaZ, iSKv, fqdc]
* Conducted three additional ablation experiments to show the effectiveness of frequency-variant encoding, indexing, and freezing the vision encoder. Reviewer [NJNg, hPaZ, fqdc]
* Added one more experiment to incorporate closed-source MLLM, gemini-2.5-flash. Reviewer [8J4Q]
* Performed an additional experiment analyzing typhoon intensity levels. Reviewer [hPaZ]
2. Added clarifications for better presentation:
* Clarified that the PIPE is the focus of this work, and incorporating additional domain variables is part of our future work. Reviewer [NJNg, hPaZ]
* Revised our writing to provide more accurate claims. Reviewer [NJNg, 8J4Q, fqdc, iSKv]

---
**We are pleased that 4 out of 5 reviewers acknowledged that we successfully addressed all their concerns.** However, while we appreciate the advice from the reviewer [iSKv], there are two concerns from the reviewer [iSKv] due to a misunderstanding. We would like to clarify the concerns.

1. Inclusion of RoPE in the Ablation Study:
* We respectfully clarified that the comparison has been incorporated in our ablation study.
* The suggestion to remove the indexing of RoPE is challenging, as indices are needed to compute the rotations.
2. Novelty.
* We respectfully emphasized that the novelty should not be judged solely on whether it appears intuitive. Intuitive methods can be highly impactful.
* This perspective aligns with the positive feedback from reviewers [NJNg, 8J4Q, hPaZ], who recognized the novel and elegant design.

Once again, we appreciate the effort invested by the AC and reviewers in providing constructive feedback.

Best regards,

Authors

---

### Decision · Program_Chairs · 2025-09-17

**Decision:**

Accept (poster)

**Comment:**

**Paper Summary:**\
This paper proposes a novel multimodal time-series forecasting framework called PIPE that incorporates visual information through a vision-language model. The paper presents extensive experiments and has been broadly recognized for its innovation. During the rebuttal phase, the authors engaged in in-depth discussions with the reviewers, leading some reviewers to improve their ratings.

**Justification:**\
Given that the reviewers unanimously recommended acceptance, and the authors engaged in multiple rounds of in-depth discussions that adequately addressed the reviewers’ concerns, I recommend accepting the paper.

**Summary of Rebuttal Period:**\
Reviewers NJNg, 8J4Q, hPaZ, and iSKv appreciated the paper’s innovation and applicability, while Reviewers NJNg and fqdc considered the experiments adequate. Reviewer 8J4Q noted that the paper was well-organized. During the rebuttal phase, the authors engaged in in-depth discussions with the reviewers and successfully addressed most of the concerns, which led to some reviewers raising their scores.